# Transport coefficients in modified Kappa distributed plasmas

**Mahmood J. Jwailes , Imad A. Barghouthi , and Qusay S. Atawnah**

Department of physics, Al-Quds university, Jerusalem, Palestine.

**Correspondence:** Mahmood J. Jwailes  (mahmood.jwailes@students.alquds.edu)

**Abstract.** This work derives transport coefficients, i.e., electrical conductivity, thermoelectric, diffusion, and mobility coefficients, for a Lorentz plasma with a modified Kappa distribution. The derivation begins by formulating transport equations (continuity, momentum, and energy) within the five-moment approximation, using the modified Kappa distribution as the zeroth-order function. Subsequently, the corresponding momentum and energy collision terms are evaluated via the Boltzmann collision integral for different types of collisions, including Coulomb collisions, hard-sphere interactions, and Maxwell molecule collisions. Next, we use the momentum equation from the five-moment approximation to obtain the generalized Ohm's law and extended Fick's law, leading to the transport coefficients. Furthermore, the influence of the kappa parameter on the collision terms and transport coefficients is analyzed. The traditional results based on the Maxwellian distribution are recovered in the limit as kappa parameter approaches infinity.

## 1   Introduction

In space plasmas, the particles' velocity distribution often deviates from the Maxwellian form. These deviations arise from processes such as wave-particle interactions, turbulence, and particle acceleration at shocks, which generate non-Maxwellian distributions. Some of these distributions exhibit non-thermal suprathermal tails that follow a power-law dependence on velocity. Such distributions are well fitted by the Kappa velocity distribution (Marsch, 2006). The Kappa distribution provides a more accurate representation of the particles' velocity distribution in non-equilibrium systems compared to the Maxwellian distribution that predicts a Gaussian function behavior of the particle velocities and fails to account for the suprathermal particles. In contrast, the Kappa distribution introduces a power-law tail that decays more slowly than the exponential tail of the Maxwellian, enabling it to accurately describe systems with significant populations of high-energy particles (Vasyliunas, 1968). This flexibility is controlled by the kappa parameter, $\kappa$, which controls the sharpness of the tail. As $\kappa$ increases, the distribution approaches the Maxwellian, while lower values of $\kappa$ emphasize the suprathermal component (Pierrard and Lazar, 2010). With $\kappa$ values usually between 2 and 6, Kappa distributions have been observed across a wide range of plasma environments, with direct measurements from several satellite missions. In the solar wind, the electron velocity distribution functions typically exhibit a thermal core, a suprathermal halo population present in all directions, and a component aligned with the interplanetary magnetic field (Pierrard et al., 2001). Kappa-distributed electrons and ions have been observed by missions such as Ulysses and Cluster. For instance, Maksimovic et al. (1997) showed that Ulysses data, fitted with Kappa functions, revealed an inverse relationship between solar wind speed and the kappa parameter $\kappa$, suggesting that suprathermal electrons influence solar wind acceleration. Similarly, Qureshi et al. (2003) also used Cluster observations to fit generalized Kappa functions to the ions' velocity distribution functions. In the Earth's magnetosheath, proton energy spectra data collected using the Heos I spacecraft confirmed that they are well fitted by Kappa distribution functions, with a kappa value around 2, particularly during crossings where no upstream waves were detected (Formisano et al., 1973). Beyond Earth, Kappa distributions have been observed in different space environments, such as Jupiter, where data from the Voyager 2 spacecraft showed that particle velocities in the planet's middle and outer magnetosphere are fitted by a Kappa distribution, with moderate kappa values indicating a transition between Maxwellian and power-law tails (Collier and Hamilton, 1995).

Given the wide range of the Kappa velocity distribution function observed in situ across planetary magnetospheres and the heliosphere and its ability to account for non-equilibrium conditions and suprathermal particles (see Pierrard and Lazar (2010), Livadiotis (2018), Davis et al. (2023), and Shizgal (2007) for more details), recent studies have investigated transport coefficients in nonequilibrium plasmas with the Kappa distribution function. In particular, Wang and Du (2017) and Ebne Abbasi et al. (2017) derived the diffusion coefficient—defined as the flux of particles due to a density gradient in the plasma—using Kappa statistics to account for suprathermal tails. Similarly, Wang and Du (2017) evaluated the mobility coefficient, which describes the particle flux under an applied electric field, within the Kappa framework. Furthermore, Du (2013) and Ebne Abbasi et al. (2017) analyzed the connection between mobility, electrical conductivity, and current density to provide a consistent description of charged-particle transport in nonequilibrium systems. In addition, Du (2013); Guo and Du (2019) calculated the thermoelectric coefficient, which links electric fields to temperature gradients and leads to the generation of electric voltages and currents, based on Kappa distributions. Finally, Du (2013); Guo and Du (2019) and Ebne Abbasi and Esfandyari-Kalejahi (2019) derived the thermal conductivity, which determines heat flux under a temperature gradient, using Kappa statistics to capture deviations from equilibrium behaviour. It is important to note that these studies primarily employed the modified Kappa distribution, which assumes a $\kappa$-independent temperature, that is, both the modified Kappa and Maxwellian distributions share the same thermal energy. This formulation makes the modified Kappa produces a stronger low-energy core and suprthermal tails compared to the Maxwellian distribution. In contrast, the standard Kappa distribution, originally introduced by Olbert (1968) and Vasyliunas (1968), is defined by $\kappa$-independent thermal speeds but $\kappa$-dependent temperatures, leading to higher-energy tails than the modified Kappa and reduced core populations lower than the Maxwellian. Distinguishing between these two forms is crucial, as the choice of distribution affects the derived transport coefficients and the physical interpretation of nonequilibrium plasma behavior, as shown by Husidic et al. (2021) which derived the electrical conductivity, thermoelectric coefficient, thermal conductivity, and diffusion and mobility coefficients for electron populations described by the standard Kappa distribution and then compared these results with those obtained for the modified Kappa distribution.

All the reviewed studies used simplified collision models rather than the full Boltzmann collision integral. The simplest models appear in Wang and Du (2017) and Ebne Abbasi and Esfandyari-Kalejahi (2019), which used Krook-type or BGK operators, offering computational simplicity but limited accuracy. More physically based models—such as those proposed by Du (2013) and Guo and Du (2019)—used macroscopic transport equations derived from idealized re-laxation assumptions. The most advanced work, presented by Ebne Abbasi et al. (2017), used the Fokker-Planck equation to model Coulomb collisions. While this captures cumulative small-angle scattering and better represents long-range Coulomb forces, it remains an approximation of the Boltzmann collision integral. Thus, all reviewed works share the same limitation: reliance on simplified collision models. Motivated by this issue, in this paper we propose a comprehensive re-evaluation of the transport coefficients based on the modified Kappa distribution, employing the Boltzmann collision integral as our collision model and adopting a more general and consistent approach through the five-moment approximation of the transport equations.

The transport equations describe the spatial and temporal evolution of the physically significant velocity moments (density, drift velocity, temperature, pressure tensor, stress tensor, and heat flow vector) derived as a practical reduction of the Boltzmann equation from a seven-dimensional partial differential equation (time plus phase space) to a set of four-dimensional equations (time and space). However, this process leads to an infinite chain of equations. To make the system solvable, a closure condition is applied by approximating the distribution function with a zeroth-order function and assuming that deviations from it are small. Different choices of the zeroth-order function have been used for closure, with the Maxwellian velocity distribution being the most common. The transport equations based on this assumption were first derived by Tanenbaum (1967) and Burgers (1969), followed by a subsequent review by Schunk (1977). These studies also obtained the collision terms—also known as transfer integrals—using the Boltzmann collision integral approach and expressed them in terms of the Chapman–Cowling collision integrals, as given in Chapman and Cowling (1990). To better capture anisotropies and departures from equilibrium, several studies have adopted more general zeroth-order forms. In particular, the bi-Maxwellian velocity distribution function—characterized by different temperatures parallel and perpendicular to the magnetic field—has been employed in several works (Demars and Schunk, 1979; Barakat and Schunk, 1981, 1982; Hellinger and Trávníček, 2009; Jubeh and Barghouthi, 2017). Beyond the pure bi-Maxwellian form, more advanced models incorporate anisotropy alongside nonthermal tails, skewness, or partial isotropization. For example, LeBlanc and Hubert (1997); Leblanc and Hubert (1998); Leblanc et al. (2000) introduced a hybrid distribution function that blends the bi-Maxwellian structure with additional functional forms to better match measured particle velocity spectra in space plasmas.

The approach used in this study involves developing a new transport theory that takes the modified Kappa distribution as the zeroth-order function, in which we derive the five-moment approximation of the transport equations and the collision terms via the Boltzmann collision integral for different types of collisions. We then relate the five-moment momentum equation to the generalized Ohm's law and the

extended Fick's law, from which the transport coefficients are obtained. The proposed methodology is implemented as follows: In Section 2, we begin with the theoretical formulation of the transport equations. Starting from Boltzmann's equation and the Boltzmann collision integral, we derive the five-moment approximation and the corresponding collision terms for the modified Kappa velocity distribution function, considering arbitrary drift velocity differences as well as temperature differences between the interacting plasma species. Next, we express the resulting collision terms in a hypergeometric representation and investigate the limiting cases where the kappa parameter approaches infinity. All these calculations cover three types of collisions: Coulomb collisions, hard-sphere interactions, and Maxwell molecules collisions. We then explore how the modified Kappa distribution influences the effective collision frequency and the thermalisation rate, providing a physical interpretation of these effects. Subsequently, we analyse the behaviour of the collision terms in the case of Coulomb collisions, focusing on how collisions affect both the momentum and the energy of the colliding particles, and how these effects differ for Maxwellian and the modified Kappa distributions. Finally, in Section 3, we derive the transport coefficients using the five-moment approximation and the obtained collision terms, followed by a discussion of how these coefficients are affected by the kappa parameter. We also provide a comparison between the derived formulas and other studies, focusing on the dependence on the kappa parameter.

## 2 Transport equations

In dealing with plasma, we describe each species in the plasma by a separate velocity distribution function $f_s(\mathbf{r}, \mathbf{v}_s, t)$, defined such that $f_s(\mathbf{r}, \mathbf{v}_s, t)\, d\mathbf{v}_s d\mathbf{r}$ represents the number of particles of species $s$, which at time $t$ have velocity between $\mathbf{v}_s$ and $\mathbf{v}_s + d\mathbf{v}_s$ and positions between $\mathbf{r}$ and $\mathbf{r} + d\mathbf{r}$. The evolution in time of the species' velocity distribution function is determined by the net effect of collisions and the flow in phase space of species under the effect of external forces. The mathematical description of this evolution is given by the Boltzmann equation (Schunk, 1977),

$$\frac{\partial f_s}{\partial t} + \mathbf{v}_s \cdot \nabla f_s + \mathbf{a}_s \cdot \nabla_{\mathbf{v}_s} f_s = \frac{\delta f_s}{\delta t}. \tag{1}$$

Here, $\nabla$ represents the gradient in coordinate space, $\nabla_{\mathbf{v}_s}$ is the gradient in velocity space, and $\mathbf{a}_s$ denotes the particle acceleration due to external forces. In most plasma applications, the main external forces acting on the charged particles are gravitational and Lorentz forces. With allowance for these forces, the acceleration becomes

$$\mathbf{a}_s = \mathbf{G} + \frac{e_s}{m_s}\left(\mathbf{E} + \frac{\mathbf{v}_s \times \mathbf{B}}{c}\right), \tag{2}$$

where $\mathbf{G}$ is the acceleration due to gravity, $e_s$ and $m_s$ are the charge and mass of species $s$, respectively, $\mathbf{E}$ is the elec-

tric field, $\mathbf{B}$ is the magnetic field, and $c$ is the speed of light. The term on the right-hand side of the Boltzmann equation, $(\delta f_s/\delta t)$, represents the rate of change of the velocity distribution function $f_s$ in a given region of phase space as a result of collisions, and its form depends on the type of collision process considered. The appropriate expression in the case of binary elastic collisions between particles (collisions governed by inverse power laws, and resonant charge exchange collisions) is the Boltzmann collision integral (Schunk, 1977; Schunk and Nagy, 2009), given by

$$\frac{\delta f_s}{\delta t} = \sum_t \int_{\mathbb{R}^3 \times \Omega} [f_s' f_t' - f_s f_t]\, \mathbf{g}_{st}\, \sigma_{st}(\mathbf{g}_{st}, \theta)\, d\Omega\, d\mathbf{v}_t, \tag{3}$$

where $d\mathbf{v}_t$ is the velocity space volume element for the target species $t$, $\mathbf{g}_{st}$ is the magnitude of the relative velocity of the colliding particles $s$ and $t$, with $\mathbf{g}_{st}$ defined as

$$\mathbf{g}_{st} = \mathbf{v}_s - \mathbf{v}_t, \tag{4}$$

$d\Omega$ is the element of solid angle in the $s$ particle reference frame, $\theta$ is the scattering angle, $\sigma_{st}(\mathbf{g}_{st}, \theta)$ is the differential scattering cross-section, defined as the number of particles scattered per solid angle $d\Omega$, per unit time, divided by the incident intensity, and the primes denote quantities evaluated after the collision. The microscopic properties of a given species $s$ can be defined and fully described by its velocity distribution function, from which, for example, the number density of particles, the zeroth-order moment, can be obtained by integrating the distribution function over the velocity space, as

$$n_s(\mathbf{r}, t) = \int_{\mathbb{R}^3} f_s(\mathbf{r}, \mathbf{v}_s, t)\, d\mathbf{v}_s, \tag{5}$$

and the drift velocity of species $s$, the first-order moment, is given by the following relation

$$\mathbf{u}_s(\mathbf{r}, t) = \langle \mathbf{v}_s \rangle, \tag{6}$$

where $\langle \mathbf{v}_s \rangle$ is the average value of $\mathbf{v}_s$, with the average value of $\xi_s(\mathbf{v}_s)$ at any position $\mathbf{r}$ and time $t$, defined as

$$\langle \xi_s \rangle = \frac{1}{n_s} \int_{\mathbb{R}^3} \xi_s(\mathbf{v}_s)\, f_s(\mathbf{r}, \mathbf{v}_s, t)\, d\mathbf{v}_s. \tag{7}$$

For higher-order moments, it is more convenient to evaluate them with respect to the drift velocity $\mathbf{u}_s$. Accordingly, Grad (1949) introduced the random velocity, defined as

$$\mathbf{c}_s(\mathbf{r}, \mathbf{v}_s, t) = \mathbf{v}_s - \mathbf{u}_s, \tag{8}$$

so that the physically significant velocity moments of the species distribution function can be written as

Temperature : $\qquad T_s(\mathbf{r}, t) = \dfrac{m_s}{3k_B} \langle c_s^2 \rangle$, $\qquad$ (9)

Pressure tensor : $\qquad \mathbf{P}_s(\mathbf{r}, t) = n_s m_s \langle \mathbf{c}_s \mathbf{c}_s \rangle$, $\qquad$ (10)

5 Stress tensor : $\qquad \boldsymbol{\tau}_s(\mathbf{r}, t) = \mathbf{P}_s - p_s \, \mathbf{I}$, $\qquad$ (11)

Heat flow vector : $\qquad \mathbf{q}_s(\mathbf{r}, t) = \dfrac{n_s m_s}{2} \langle c_s^2 \, \mathbf{c}_s \rangle$, $\qquad$ (12)

where $k_B$ is the Boltzmann constant, $\mathbf{I}$ is a unit dyadic, $p_s$ is the partial pressure and defined as

$$p_s = n_s k_B T_s. \qquad (13)$$

10 The starting point for deriving the transport equations is the Boltzmann equation. These equations can be obtained by multiplying the Boltzmann equation by an appropriate function of velocity and then integrating over the velocity space. In particular, multiplying equation (1) by the factors $1$, $m_s \mathbf{c}_s$, 15 $\frac{1}{2} m_s c_s^2$, $m_s \mathbf{c}_s \mathbf{c}_s$, and $\frac{1}{2} m_s c_s^2 \mathbf{c}_s$, followed by integration over the velocity space, gives the continuity, momentum, energy, pressure tensor, and heat flow equations, respectively. Together, these form the general transport equations for species $s$, as presented in Schunk (1977), Schunk and Nagy (2009), 20 and Bittencourt (2004). The general transport equations do not constitute a closed system because the equation governing the moment of order $l$ contains the moment of order $l+1$. That is, the continuity equation describes the evolution of the density, but it also contains the drift velocity, and similar de-25 pendencies occur in the higher order moment equations. To close the system, it is necessary to adopt an approximate expression for the velocity distribution function $f_s$. A common mathematical technique can be used to do that, is expanding $f_s$ in a complete orthogonal series of the form (Grad, 1949; 30 Mintzer, 1965),

$$f_s(\mathbf{r}, \mathbf{c}_s, t) = f_s^{(0)}(\mathbf{r}, \mathbf{c}_s, t) \sum_i a_i(\mathbf{r}, t) M_i(\mathbf{r}, \mathbf{c}_s, t), \qquad (14)$$

where $f_s^{(0)}$ is an appropriate zeroth-order velocity distribution function, $M_i$ represents a complete set of orthogonal polynomials, and $a_i$ are the unknown expansion coefficients. 35 The zeroth-order distribution function and the orthogonal set of polynomials are generally chosen so that the series converges rapidly, meaning that only a few terms in the series expansion are needed to describe the distribution function. Different levels of approximation are possible, depending on 40 the number of terms retained in the series expansion.

## 2.1 The five-moment approximation

The first term in the series expansion of equation (14) is 1, regardless of which zeroth-order distribution function $f_s^{(0)}$

is chosen (Mintzer, 1965). Therefore, assuming the species distribution function $f_s$ is represented by the first term of the 45 expansion, we have

$$f_s(\mathbf{r}, \mathbf{c}_s, t) = f_s^{(0)}(\mathbf{r}, \mathbf{c}_s, t). \qquad (15)$$

The approximation in equation (15) reduces the general system of transport equations to just the continuity, momentum, and energy equations for each species $s$, 50

$$\frac{\delta n_s}{\delta t} = \frac{\partial n_s}{\partial t} + \nabla \cdot (n_s \mathbf{u}_s), \qquad (16)$$

$$\frac{\delta \mathbf{M}_s}{\delta t} = n_s m_s \frac{\mathrm{D}_s \mathbf{u}_s}{\mathrm{D} t} + \nabla \cdot \mathbf{P}_s$$
$$\qquad - n_s m_s \mathbf{G} - n_s e_s \left( \mathbf{E} + \frac{\mathbf{u}_s \times \mathbf{B}}{c} \right), \qquad (17)$$

$$\frac{\delta E_s}{\delta t} = \frac{3}{2} \frac{\mathrm{D}_s p_s}{\mathrm{D} t} + \frac{3}{2} p_s (\nabla \cdot \mathbf{u}_s) + \nabla \cdot \mathbf{q}_s + \mathbf{P}_s : \nabla \mathbf{u}_s, \quad (18)$$

where the operation $\mathbf{P}_s : \nabla \mathbf{u}_s$ corresponds to the double 55 product of the two tensors $\mathbf{P}_s$ and $\nabla \mathbf{u}_s$, and the operator $\mathrm{D}_s / \mathrm{D} t$ is defined as

$$\frac{\mathrm{D}_s}{\mathrm{D} t} = \frac{\partial}{\partial t} + \mathbf{u}_s \cdot \nabla. \qquad (19)$$

The set of equations (16−18) was initially derived with no assumption about the zeroth-order function $f_s^{(0)}$ (Tanen-60 baum, 1967). In the present study, we adopt the drifting modified Kappa distribution (MK) as the zeroth-order function. The drifting modified Kappa distribution is commonly written in the following form (Livadiotis, 2018; Davis et al., 2023), 65

$$f_s^{(0)} = f_s^{\mathrm{MK}}(\mathbf{r}, \mathbf{c}_s, t) = \frac{n_s \eta(\kappa_s)}{\pi^{3/2} \, w_s^3} \left( 1 + \frac{c_s^2}{\kappa_{0_s} w_s^2} \right)^{-\kappa_s - 1}, \quad (20)$$

where the thermal velocity of species $s$, denoted by $w_s$, is given by

$$w_s = \sqrt{\frac{2k_B T_s}{m_s}}, \qquad (21)$$

with $m_s$ and $T_s$ denoting the mass and the absolute tempera-70 ture of species $s$ respectively, and $k_B$ is the Boltzmann constant. The function $\eta(\kappa_s)$, which depends on the kappa parameter $\kappa_s$, is defined as

$$\eta(\kappa_s) = \kappa_{0s}^{-3/2} \frac{\Gamma(\kappa_s + 1)}{\Gamma(\kappa_s - 1/2)}, \quad \kappa_{0_s} = \kappa_s - \frac{3}{2}. \qquad (22)$$

Here, $\kappa_{0_s}$ is the invariant kappa index, and $\kappa_s$ is a shape pa-75 rameter that controls the power-law tails, sometimes referred to as the spectral index, with the condition $\kappa_s > 3/2$. This condition prevents the modified Kappa distribution function in equation (20) from collapsing (Pierrard and Lazar, 2010). If the chosen zeroth-order distribution function $f_s^{(0)}$ satisfies 80

$$\mathbf{q}_s = \boldsymbol{\tau}_s = 0, \qquad (23)$$

as in the drifting Maxwellian distribution and the drifting modified Kappa distribution (Scherer et al., 2019), we can write equations (16−18) as (Schunk, 1977),

$$\frac{\delta n_s}{\delta t} = \frac{\partial n_s}{\partial t} + \nabla \cdot (n_s \mathbf{u}_s), \tag{24}$$

$$\frac{\delta \mathbf{M}_s}{\delta t} = n_s m_s \frac{D_s \mathbf{u}_s}{Dt} + \nabla p_s$$

$$- n_s m_s \mathbf{G} - n_s e_s \left( \mathbf{E} + \frac{\mathbf{u}_s \times \mathbf{B}}{c} \right), \tag{25}$$

$$\frac{\delta E_s}{\delta t} = \frac{3}{2} \frac{D_s p_s}{Dt} + \frac{5}{2} p_s (\nabla \cdot \mathbf{u}_s). \tag{26}$$

These equations are known as the five-moment approximation of the transport equations because each species is characterized by five parameters: density, three components of drift velocity, and temperature. At this level of approximation, stress, heat flow, and all higher-order velocity moments are neglected, and the species' properties are expressed in terms of density, drift velocity, and temperature.

## 2.2 Collision terms

The terms appearing on the left-hand side of the five-moment approximation, equations (24−26), are called the collision terms, also known as the transfer collision integral. These terms represent the moments of the Boltzmann collision integral and describe the rate of change of density, momentum, and energy due to collisions, and they are defined as follows

$$\frac{\delta n_s}{\delta t} = \int_{\mathbb{R}^3} \frac{\delta f_s}{\delta t} \, d\mathbf{c}_s, \tag{27}$$

$$\frac{\delta \mathbf{M}_s}{\delta t} = m_s \int_{\mathbb{R}^3} \mathbf{c}_s \frac{\delta f_s}{\delta t} \, d\mathbf{c}_s, \tag{28}$$

$$\frac{\delta E_s}{\delta t} = \frac{m_s}{2} \int_{\mathbb{R}^3} c_s^2 \frac{\delta f_s}{\delta t} \, d\mathbf{c}_s, \tag{29}$$

where the Boltzmann's collision integral, expressed in terms of the random velocities $\mathbf{c}_s$ and $\mathbf{c}_t$, takes the form:

$$\frac{\delta f_s}{\delta t} = \sum_t \int_{\mathbb{R}^3 \times \Omega} [f_s' f_t' - f_s f_t] \, g_{st} \, \sigma_{st}(g_{st}, \theta) \, d\Omega \, d\mathbf{c}_t, \tag{30}$$

with the functions $f_s'$, $f_t'$, $f_s$, and $f_t$ depend on $(\mathbf{r}, t, \mathbf{c}_s \text{ or } \mathbf{c}_t)$. Calculating the collision terms involves solving the integrals appearing in equations (27-29). The process begins by substituting the Boltzmann collision integral from equation (30) and rewriting the resulting integrals in an equivalent form that does not require the distribution functions after the collision, $f_s' f_t'$. We then use the momentum transfer cross-section integral, defined as

$$Q_{st}^{(1)} (g_{st}) = \int_{\Omega} (1 - \cos\theta) \, \sigma_{st} (g_{st}, \theta) \, d\Omega, \tag{31}$$

to write the collision terms as (Schunk and Nagy, 2009),

$$\frac{\delta n_s}{\delta t} = 0, \tag{32}$$

$$\frac{\delta \mathbf{M}_s}{\delta t} = -\sum_t m_{st} \int_{\mathbb{R}^3 \times \mathbb{R}^3} f_s f_t \, g_{st} \, Q_{st}^{(1)} \, \mathbf{g}_{st} \, d\mathbf{c}_t d\mathbf{c}_s, \tag{33}$$

$$\frac{\delta E_s}{\delta t} = -\sum_t m_{st} \int_{\mathbb{R}^3 \times \mathbb{R}^3} f_s f_t \, g_{st} \, Q_{st}^{(1)} \left( \hat{\mathbf{V}}_c \cdot \mathbf{g}_{st} \right) d\mathbf{c}_t \, d\mathbf{c}_s, \tag{34}$$

where the dot product is written as

$$\hat{\mathbf{V}}_c \cdot \mathbf{g}_{st} = \frac{m_s \mathbf{c}_s \cdot \mathbf{g}_{st} + m_t \mathbf{c}_t \cdot \mathbf{g}_{st} + m_t \Delta\mathbf{u} \cdot \mathbf{g}_{st}}{m_s + m_t}, \tag{35}$$

and the reduced mass $m_{st}$ is expressed as

$$m_{st} = \frac{m_s m_t}{m_s + m_t}. \tag{36}$$

Equations (33) and (34) are expressed in terms of the momentum transfer cross-section integral $Q_{st}^{(1)}$, which depends explicitly on the nature of the particles' interaction; different collision models yield different functional forms for the cross-section. In the present work, we examine three distinct cases: Coulomb collisions, hard-sphere interactions, and Maxwell molecule collisions. The momentum transfer cross-section for Coulomb collisions is given by

$$Q_{st}^{(1)} (g_{st}) = \frac{Q_{Co}}{g_{st}^4}, \quad Q_{Co} = 4\pi \left( \frac{e_s e_t}{4\pi \varepsilon_0 m_{st}} \right)^2 \ln\Lambda, \tag{37}$$

where $e_s$ and $e_t$ are the charges of species $s$ and $t$, respectively, $\varepsilon_0$ is the permittivity of free space, and $\ln\Lambda$ is the Coulomb logarithm. For hard-sphere interactions, the momentum transfer cross-section,

$$Q_{st}^{(1)} = Q_{HS} = \pi \sigma^2, \tag{38}$$

is a constant ($\sigma$ is the sum of the radii of the colliding particles). In the case of Maxwell molecule collisions, the momentum transfer cross-section is

$$Q_{st}^{(1)} = \frac{Q_{MC}}{g_{st}}, \quad Q_{MC} = 0.844\pi \left( \frac{K_{st}}{m_{st}} \right)^{1/2}, \tag{39}$$

where $K_{st}$ denotes a proportionality constant that measures the force magnitude between particles. Now, we proceed to calculate the momentum and energy collision terms, given in equations (33) and (34), for the five-moment approximation, under the assumption that the velocity distribution function of both interacting species $s$ and $t$ is a drifting modified Kappa distribution. The general expressions for the collision terms are summarized below, while detailed derivations for

the three types of collisions are provided in Appendix A.

$$\frac{\delta n_s}{\delta t} = 0, \tag{40}$$

$$\frac{\delta \mathbf{M}_s}{\delta t} = \sum_t n_s m_s \, \nu_{st}^{\mathrm{MK}}(\kappa_s, \kappa_t) \, \Phi(\varepsilon_{st}) \, \Delta \mathbf{u}_{st}, \tag{41}$$

$$\frac{\delta E_s}{\delta t} = \sum_t n_s \left[ \frac{3}{2} k_B \, \nu_{st,T}^{\mathrm{MK}}(\kappa_s, \kappa_t) \, \Psi(\varepsilon_{st}) \, \Delta \mathrm{T}_{st}^{\mathrm{MK}} \right.$$
$$\left. + m_{st} \, \nu_{st}^{\mathrm{MK}}(\kappa_s, \kappa_t) \, \Phi(\varepsilon_{st}) \, |\Delta \mathbf{u}_{st}|^2 \right], \tag{42}$$

where the relative drift velocity $\Delta \mathbf{u}_{st}$ and relative temperature difference $\Delta \mathrm{T}_{st}^{\mathrm{MK}}$ are defined by

$$\Delta \mathbf{u}_{st} = \mathbf{u}_t - \mathbf{u}_s, \tag{43}$$

$$\Delta \mathrm{T}_{st}^{\mathrm{MK}} = \mathrm{H}(\kappa_t) T_t - \mathrm{H}(\kappa_s) T_s, \tag{44}$$

and the drift-to-thermal speed ratio $\varepsilon_{st}$ is given by

$$\varepsilon_{st} = \frac{|\Delta \mathbf{u}_{st}|}{w_{st}}, \quad w_{st} = \sqrt{\frac{2 k_B T_{st}}{m_{st}}}, \tag{45}$$

with the reduced mass $m_{st}$ given in equation (36) and the reduced temperature $T_{st}$ defined by

$$T_{st} = \frac{m_s T_t + m_t T_s}{m_s + m_t}. \tag{46}$$

The kappa-dependent terms $\nu_{st}^{\mathrm{MK}}$ and $\nu_{st,T}^{\mathrm{MK}}$ represent, respectively, the effective collision frequency and the thermal equilibration rate (or simply the thermalisation rate) for systems described by the modified Kappa distribution, and they are defined as

$$\nu_{st}^{\mathrm{MK}}(\kappa_s, \kappa_t) = \nu_{st} \, \mathrm{D}(\kappa_s, \kappa_t), \tag{47}$$

$$\nu_{st,T}^{\mathrm{MK}}(\kappa_s, \kappa_t) = 2 \, \frac{m_{st}}{m_t} \, \nu_{st}^{\mathrm{MK}}, \tag{48}$$

where $\nu_{st}$ denote the effective collision frequency rate for systems governed by the Maxwellian distribution. The factors $\nu_{st}, \Phi, \Psi, \mathrm{D},$ and $\mathrm{H}$ forms change depending on the type of collision, such as Coulomb, hard-sphere, or Maxwell molecule collisions, and can be summarized as follows:

*Coulomb collisions*:

The effective collision frequency for Coulomb collisions in the Maxwellian case is

$$\nu_{st} = \nu_{st}^{\mathrm{Co}} = \frac{4}{3} \frac{n_t}{\pi^{1/2}} \frac{m_t}{m_s + m_t} \left( \frac{1}{2 k_B} \frac{m_{st}}{T_{st}} \right)^{3/2} \mathrm{Q}_{\mathrm{Co}}, \tag{49}$$

where $\mathrm{Q}_{\mathrm{Co}}$ is defined in equation (37). The functions $\Phi$ and $\Psi$ are given by

$$\Phi = \Phi_{\mathrm{Co}}(\varepsilon_{st}) = \frac{3\sqrt{\pi}}{4} \frac{\mathrm{erf}(\varepsilon_{st})}{\varepsilon_{st}^3} - \frac{3 e^{-\varepsilon_{st}^2}}{2 \varepsilon_{st}^2}, \tag{50}$$

$$\Psi = \Psi_{\mathrm{Co}}(\varepsilon_{st}) = e^{-\varepsilon_{st}^2}. \tag{51}$$

The kappa-dependent factors D and H are defined as

$$\mathrm{D}(\kappa_s, \kappa_t) = \frac{(\kappa_s - 1/2)}{(\kappa_s - 3/2)} \frac{(\kappa_t - 1/2)}{(\kappa_t - 3/2)}, \tag{52}$$

$$\mathrm{H}(\kappa_\alpha) = \frac{\Gamma(\kappa_\alpha)}{\Gamma(\kappa_\alpha + 1/2)} (\kappa_\alpha - 3/2)^{1/2}, \quad \alpha = s, t. \tag{53}$$

*Hard-sphere interactions*:

The effective collision frequency for Hard-sphere in the Maxwellian case is

$$\nu_{st} = \nu_{st}^{\mathrm{HS}} = \frac{8}{3} \frac{n_t}{\pi^{1/2}} \frac{m_t}{m_s + m_t} \left( 2 k_B \frac{T_{st}}{m_{st}} \right)^{1/2} \mathrm{Q}_{\mathrm{HS}}, \tag{54}$$

where $\mathrm{Q}_{\mathrm{HS}}$ is defined in equation (38). The functions $\Phi$ and $\Psi$ are given by

$$\Phi = \Phi_{\mathrm{HS}}(\varepsilon_{st}) = \frac{3}{8} \left( 1 + \frac{1}{2\varepsilon_{st}^2} \right) e^{-\varepsilon_{st}^2}$$
$$+ \frac{3\sqrt{\pi}}{8} \left( \varepsilon_{st} + \frac{1}{\varepsilon_{st}} - \frac{1}{4\varepsilon_{st}^3} \right) \mathrm{erf}(\varepsilon_{st}), \tag{55}$$

$$\Psi = \Psi_{\mathrm{HS}}(\varepsilon_{st}) = \frac{\sqrt{\pi}}{2} \left( \varepsilon_{st} + \frac{1}{2\varepsilon_{st}} \right) \mathrm{erf}(\varepsilon_{st}) + \frac{e^{-\varepsilon_{st}^2}}{2}. \tag{56}$$

The kappa-dependent factors D and H are defined the same as in equation (52) and (53).

*Maxwell molecule collisions*

The effective collision frequency for Maxwell molecule collisions in the Maxwellian case is

$$\nu_{st} = \nu_{st}^{\mathrm{MC}} = \frac{n_t m_t}{m_s + m_t} \mathrm{Q}_{\mathrm{MC}}, \tag{57}$$

where $\mathrm{Q}_{\mathrm{MC}}$ is defined in (39). The functions $\Phi$ and $\Psi$ are given by

$$\Phi = \Phi_{\mathrm{MC}}(\varepsilon_{st}) = 1, \quad \Psi = \Psi_{\mathrm{MC}}(\varepsilon_{st}) = 1, \tag{58}$$

The factors D and H are defined as

$$\mathrm{D}(\kappa_s, \kappa_t) = 1, \quad \mathrm{H}(\kappa_\alpha) = 1, \quad \alpha = s, t. \tag{59}$$

The collision terms can be derived for non-drifting modified Kappa distributions by setting the drift velocities of both interacting particles $s$ and $t$ to zero, $\mathbf{u}_s = \mathbf{u}_t = 0$, which gives $\Delta \mathbf{u}_{st} = 0$ and $\Phi(0) = \Psi(0) = 1$, in equations (40-42). The same expressions are obtained when the drift velocities of species $s$ and $t$ are equal, i.e., $\mathbf{u}_s = \mathbf{u}_t$.

### 2.2.1 Hypergeometric representation

The resulting collision terms in case of Coulomb collision and hard sphere interaction can be written in terms of the hypergeometric functions. This is done by expressing the $\Phi$'s

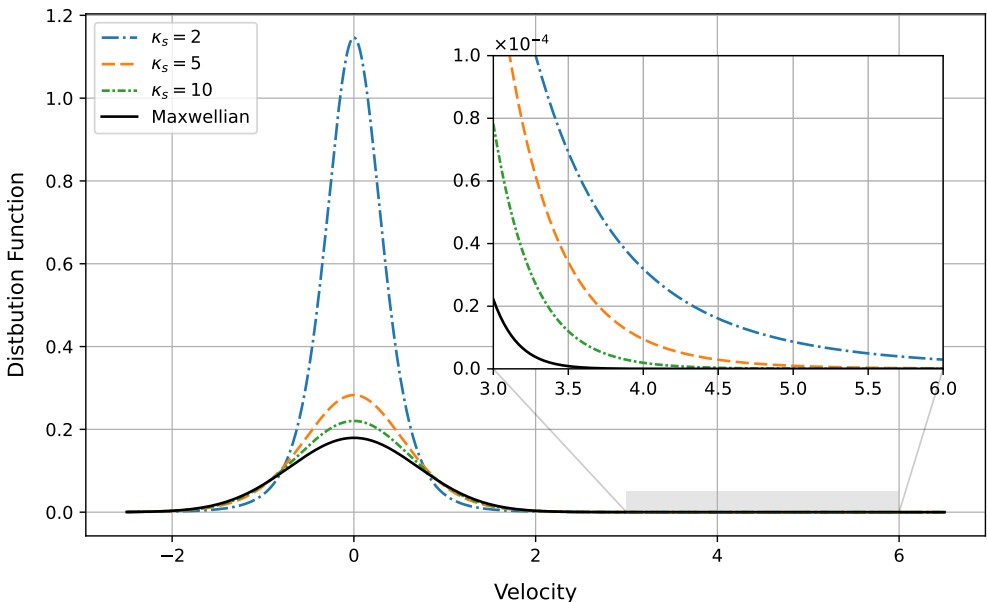

**Figure 1.** A schematic comparison between modified Kappa velocity distributions for $\kappa_s$ values 2, 5, and 10, and the Maxwellian velocity distribution.

and $\Psi$'s, equations (50, 51, 55, and 56), in the hypergeometric representation, such that

*Coulomb collisions*:

$$\Phi_{\mathrm{Co}}(\varepsilon_{st}) = {}_1F_1\left(\frac{3}{2};\frac{5}{2};-\varepsilon_{st}^2\right), \tag{60}$$

$$\Psi_{\mathrm{Co}}(\varepsilon_{st}) = {}_1F_1\left(\frac{1}{2};\frac{1}{2};-\varepsilon_{st}^2\right). \tag{61}$$

*Hard-sphere interactions*:

$$\Phi_{\mathrm{HS}}(\varepsilon_{st}) = {}_1F_1\left(-\frac{1}{2};\frac{5}{2};-\varepsilon_{st}^2\right), \tag{62}$$

$$\Psi_{\mathrm{HS}}(\varepsilon_{st}) = {}_1F_1\left(-\frac{1}{2};\frac{3}{2};-\varepsilon_{st}^2\right). \tag{63}$$

#### 2.2.2 Limiting case: kappa parameter approaches infinity

One of the special properties of the modified Kappa distribution is that, as $\kappa_s$ approaches infinity, the modified Kappa distribution reduces to a Maxwellian distribution (Pierrard and Lazar, 2010). Specifically, we have

$$\lim_{\kappa_s \to \infty} \eta(\kappa_s) = 1 \tag{64}$$

and

$$\lim_{\kappa_s \to \infty}\left[1 + \frac{a}{(\kappa_s - 3/2)}\right]^{-\kappa_s - 1} = e^{-a}, \tag{65}$$

Therefore, the modified Kappa distribution becomes identically Maxwellian distribution

$$\lim_{\kappa_s \to \infty} f_s^{\mathrm{MK}} = \frac{n_s}{\pi^{3/2}\,w_s^3}\exp\left(-\frac{c_s^2}{w_s^2}\right), \tag{66}$$

as shown in figure 1, where as $\kappa_s$ gets larger, the distribution becomes closer and closer to the Maxwellian distribution. This provides further confirmation that the derived formulas are correct by taking the limit of the collision terms as $\kappa \to \infty$, $\kappa = \kappa_s = \kappa_t$, and comparing the resulting limits with existing formulas for the Maxwellian. The collision terms given in equations (40–42) depend on the kappa parameter through the effective collision frequency, the thermalisation rate, and the relative temperature difference, $\nu_{st}^{\mathrm{MK}}$, $\nu_{st,T}^{\mathrm{MK}}$, and $\Delta\mathrm{T}_{st}^{\mathrm{MK}}$, respectively. These quantities are expressed using the two functions $\mathrm{D}(\kappa_s, \kappa_t)$ and $\mathrm{H}(\kappa_\alpha)$, $\alpha = s, t$. Consequently, the limits of the collision terms reduce to the following limits

$$\lim_{\kappa \to \infty} \nu_{st}^{\mathrm{MK}}(\kappa, \kappa) = \nu_{st}, \tag{67}$$

$$\lim_{\kappa \to \infty} \nu_{st,T}^{\mathrm{MK}}(\kappa, \kappa) = \nu_{st,T}, \tag{68}$$

$$\lim_{\kappa \to \infty} \Delta\mathrm{T}_{st}^{\mathrm{MK}} = T_t - T_s = \Delta\mathrm{T}_{st}. \tag{69}$$

Hence,

$$\lim_{\kappa \to \infty} \mathrm{D}(\kappa, \kappa) = \lim_{\kappa \to \infty} \mathrm{H}(\kappa) = 1. \tag{70}$$

With $\nu_{st,T}$ denoting the thermalisation rate for systems governed by the Maxwellian distribution, defined as

$$\nu_{st,T} = 2\frac{m_{st}}{m_t}\nu_{st}. \tag{71}$$

Therefore, in the limit $\kappa$ approaches $\infty$, the collision terms (40–42) recover the form

$$\lim_{\kappa\to\infty}\left[\frac{\delta n_s}{\delta t}\right] = 0, \tag{72}$$

$$\lim_{\kappa\to\infty}\left[\frac{\delta \mathbf{M}_s}{\delta t}\right] = \sum_t n_s m_s \nu_{st}\,\Phi(\varepsilon_{st})\,\Delta\mathbf{u}_{st}, \tag{73}$$

$$\lim_{\kappa\to\infty}\left[\frac{\delta E_s}{\delta t}\right] = \sum_t \frac{n_s m_s \nu_{st}}{m_s + m_t}\left[3k_B\,\Psi(\varepsilon_{st})\,\Delta\mathrm{T}_{st}\right.$$
$$\left. + m_t\,\Phi(\varepsilon_{st})\,|\Delta\mathbf{u}_{st}|^2\right]. \tag{74}$$

The resulting limits gives exactly the same results as the Maxwellian distribution (Schunk and Nagy, 2009), with the same definitions of $\Phi, \Psi$, and $\nu_{st}$.

### 2.3 Effective collision frequency and thermalisation rate in systems with modified Kappa distributions

Within the framework of the five-moment approximation of the transport equations, the effective collision frequency and the thermalisation rate can be obtained directly from the momentum and energy collision terms. As derived in the previous section for the modified Kappa distribution, equations (47) and (48) present the effective collision frequency and the thermalisation rate, which are essential for understanding the exchange of momentum and energy between particles due to collisions. The effective collision frequency reflects the average rate of how frequently collisions occur, determining the efficiency of momentum transfer within the system, while the thermalisation rate measures how rapidly the system approaches thermal equilibrium through collisions. Equations (47) and (48) show that the modified Kappa distribution affects the effective collision frequency and the thermalisation rate through the kappa dependent term $\mathrm{D}(\kappa_s, \kappa_t)$. This factor depends explicitly on the kappa parameters $\kappa_s$ and $\kappa_t$ of the interacting species $s$ and $t$, and its form changes depending on the type of collisions under consideration. To compare the effective collision frequency and thermalisation rate of the modified Kappa and Maxwellian distributions, we must first understand how their particle velocity distributions differ. While the Maxwellian distribution has most particles concentrated around the distribution core, with low and intermediate velocity magnitudes, the modified Kappa distribution shifts this balance by increasing the number of particles both in the low-energy core and in the high-energy tails, (i.e., at low and high velocity magnitudes), as shown in Figure (1). This redistribution in the particle velocities is directly related to the effective collision frequency and the

thermalisation rate. In collision processes, such as Maxwell molecule collisions, where the collision frequency does not depend on particle velocity, the redistribution has no effect. The effective collision frequency and thermalisation rate remain the same even when modified Kappa distribution is used. This is confirmed by the result $\mathrm{D} = 1$, which shows that the kappa parameter does not change either the effective collision frequency or the thermalisation rate compared with the Maxwellian case,

$$\nu_{st}^{\mathrm{MK}} = \nu_{st}, \quad \text{and} \quad \nu_{st,T}^{\mathrm{MK}} = \nu_{st,T}. \tag{75}$$

In contrast, when collisions strongly depend on particle velocity, the modified Kappa distribution significantly affects both the effective collision frequency and the thermalisation rate. This effect becomes particularly evident in processes such as Coulomb collisions and hard-sphere interactions, where the velocity distribution strongly shapes the interaction dynamics. In these cases, the functions D vary according to the kappa parameters $\kappa_s$ and $\kappa_t$, as given in equations (52). To compare the effective collision frequency and thermalisation rate with the Maxwellian case, and to better understand their behaviour, we consider the special case $\kappa = \kappa_s = \kappa_t$, so that the expressions, $\nu_{st}^{\mathrm{MK}}$ and $\nu_{st,T}^{\mathrm{MK}}$, reduce to

$$\nu_{st}^{\mathrm{MK}} = \nu_{st}\left(\frac{\kappa - 1/2}{\kappa - 3/2}\right)^2, \tag{76}$$

$$\nu_{st,T}^{\mathrm{MK}} = 2\frac{m_{st}}{m_t}\nu_{st}^{\mathrm{MK}}, \tag{77}$$

The effective collision frequency for the modified Kappa distribution in equation (76) agrees exactly with Livadiotis (2019) in the $\kappa$ dependency, where both share the same functional form. Equations (76) and (77) show that for small values of $\kappa$, both the effective collision frequency and the thermalisation rate are large and decreasing as $\kappa$ increases. As $\kappa$ goes to infinity, the kappa term in equation (76) approaches 1, and the results converge to those of the Maxwellian distribution, as illustrated in Figure 2. In this figure, we have plotted the $\kappa$ dependency for both the effective collision frequency and the thermalisation rate; in other words, the ratios $\nu_{st}^{\mathrm{MK}}/\nu_{st}$ and $\nu_{st,T}^{\mathrm{MK}}/\nu_{st,T}$ as functions of the kappa parameter. This behaviour arises from the redistribution of the particle velocities in the modified Kappa distribution, where at low values of $\kappa$, the number of particles near the core with a small velocity magnitudes is very high compared to the Maxwellian distribution. These particles increase the collision frequency in case of Coulomb collisions and Maxwell molecules interactions, since these interactions are inversely proportional to velocity, making the effective collision frequency and the thermalisation rate are enhanced at low kappa values.

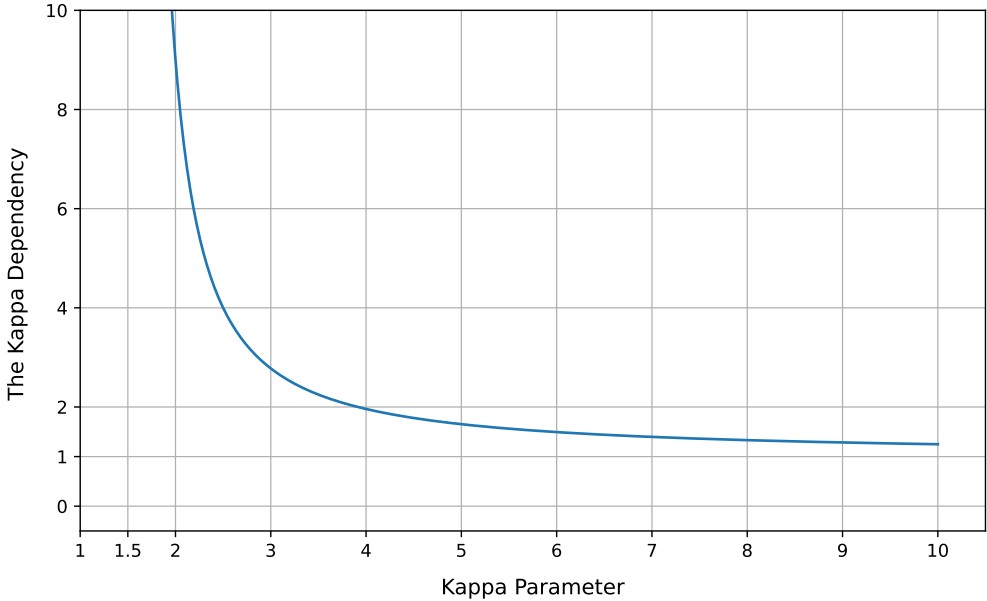

**Figure 2.** The kappa dependency for both the effective collision frequency and the thermalisation rate.

## 2.4 Variations of collision terms as a result of Coulomb collisions

The collision terms for the five-moment approximation, presented in equations (40−42), describe how density, momentum, and energy, for particles $s$ change under the effect of collisions. These terms depend on three variables, number density $n_s$, drift velocity $\mathbf{u}_s$, and temperature $T_s$ for particles $s$, as well as on the parameters of particles $t$, number density $n_t$, drift velocity $\mathbf{u}_t$, and temperature $T_t$. Additionally, two functions of $\kappa_s$ and $\kappa_t$, $\mathrm{D}(\kappa_s, \kappa_t)$ and $\mathrm{H}(\kappa_\alpha), \alpha = s, t$, contribute to the effective collision frequency, the thermalisation rate and the relative temperature difference. The masses of both interacting particles $s$ and $t$, $(m_s, m_t)$, are constant and remain unchanged throughout the collision process for all types of collisions, this makes the density coefficient to be zero according to equation (40). In this section, we examine how the collision terms, in the case of Coulomb collisions, vary with respect to the variables $(n_s, \mathbf{u}_s, T_s)$, and the parameters $(n_t, \mathbf{u}_t, T_t)$, and we compare the results for the modified Kappa and Maxwellian distributions.

### 2.4.1 Maxwellian distribution

In the Maxwellian case, both functions $\mathrm{D}(\kappa_s, \kappa_t)$ and $\mathrm{H}(\kappa_\alpha), \alpha = s, t$ are set to one, see Sub-subsection 2.2.2. Consequently, the effective collision frequency, the thermalisation rate, and the relative temperature difference in the collision terms simplify to

$$\nu_{st}^{\mathrm{MK}} = \nu_{st}, \quad \nu_{st,T}^{\mathrm{MK}} = \nu_{st,T}, \quad \text{and} \quad \Delta\mathrm{T}_{st}^{\mathrm{MK}} = \Delta\mathrm{T}_{st}. \tag{78}$$

For clarity, we will discuss the behaviour of the collision terms in three cases, each involving a particular choice of variables or parameters.

**First case**, the *number density* of the interacting particles, $n_s$ and $n_t$. From equations (40−42), $n_s$ and $n_t$ appears as a product, indicating that an increase in the number density of either particle $s$ or $t$ will increase the influence of collisions on both the momentum and energy of the $s$ particles. Such a result is reasonable because the number density measures particles per unit volume—more particles in the same volume lead to more collisions between $s$ and $t$, causing greater changes in momentum and energy due to collisions.

**Second case**, the *drift velocities* of the interacting particles $\mathbf{u}_s, \mathbf{u}_t$, and the *temperature* of the $s$ particles $T_s$. Equations (40−42) show that the collision terms depend on the difference in drift velocity, $\Delta\mathbf{u}_{st} = \mathbf{u}_t - \mathbf{u}_s$, and on $T_s$. Figures (3a) and (3c) display the isolines of the momentum and energy collision terms as functions of $\Delta\mathbf{u}_{st}$ and $T_s$, with all other constants set to 1.0 for simplicity. Assuming identical parameters for all $t$ particles, the summation over $t$, in equations (40−42), reduces to multiplication by their number, $N_t$, which is set to 1000 for easier comparison with other cases. Figure (3a) shows the magnitude of the momentum collision term, assuming that the direction of $\Delta\mathbf{u}_{st}$ is along the $z$-axis. The figure (3a) indicates that the momentum of the particles $s$ remains unchanged when the difference in drift velocities, $\Delta\mathbf{u}_{st}$, is zero, regardless of the temperature $T_s$. This means that the drift velocities of the two particles are equal or both zero, so they are not moving relative to each other, or the system has no net current flow (e.g., no applied electric field).

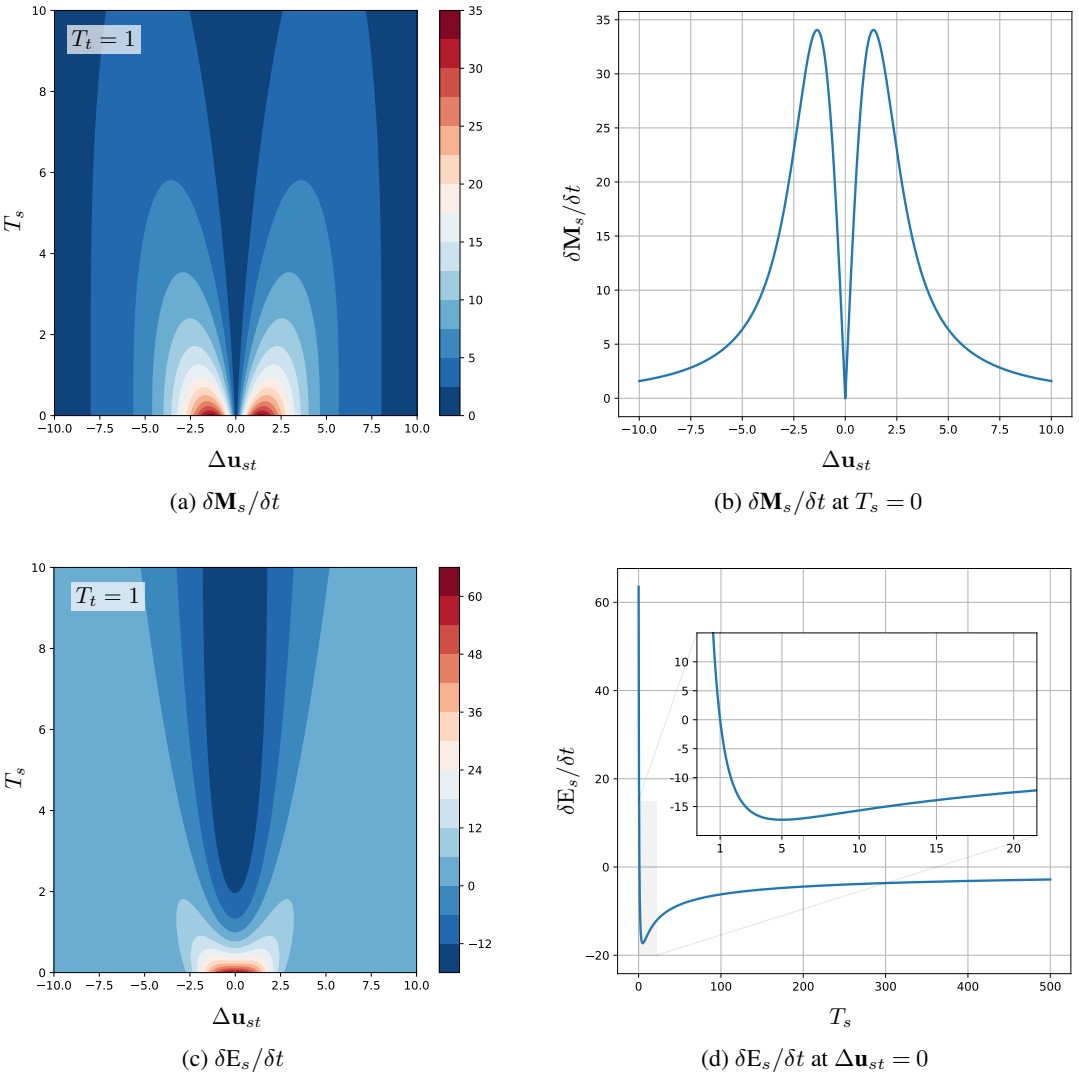

**Figure 3.** The momentum and energy collision terms for the Maxwellian velocity distribution function in the case of Coulomb collisions.

Under these conditions, Coulomb collisions can be treated as elastic collisions, so that the total kinetic energy and momentum of the system are conserved, resulting in no change in momentum caused by the collisions. To see how the momentum collision term changes relative to $\Delta\mathbf{u}_{st}$, we plot its cross-section at $T_s = 0$, as shown in Figure (3b). When one particle's drift velocity is slightly larger than the other's, the momentum change of particle $s$ due to collisions increases until reaching its maximum. Beyond this maximum point, as the absolute value of the difference between the drift velocities becomes very large (i.e., $\mathbf{u}_s \gg \mathbf{u}_t$ or $\mathbf{u}_t \gg \mathbf{u}_s$), collisions have less effect on the momentum. In the limit of very large $\Delta\mathbf{u}_{st}$, the momentum collision term approaches zero because particles moving at significantly different speeds are more likely to pass each other without interacting, i.e. reducing the number of collisions. Referring back to Figure (3a),

as the temperature of the $s$ particles increases, the impact of collisions on their momentum decreases until it eventually vanishes. This occurs because the effective collision frequency, $\nu_{st}$, for Coulomb collisions in the Maxwellian case, defined in equation (49), between the particles decreases with increasing temperature as

$$\nu_{st} = \nu_{st}^{\text{Co}} \propto 1/T_{st}^{3/2}, \tag{79}$$

showing that collisions become less probable at higher temperatures, leading to a reduction in their influence on momentum. Figure (3c) represents the energy collision term. The graph shows that maximum energy exchange occurs when the difference in the drift velocities is zero. As mentioned earlier, in this case, both momentum and kinetic energy are conserved, meaning there is no change in these quantities.

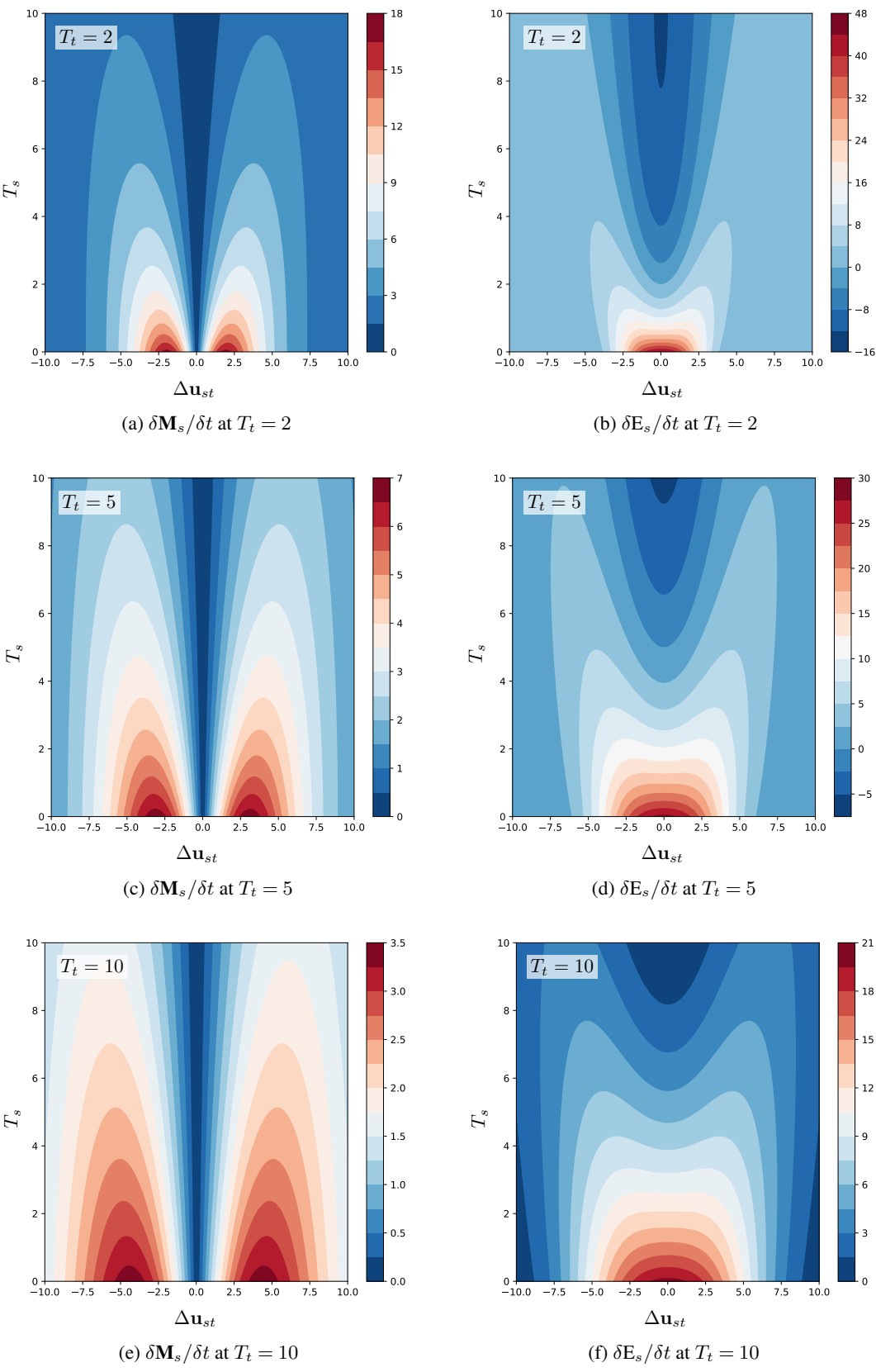

(a) $\delta \mathbf{M}_s/\delta t$ at $T_t = 2$

(b) $\delta \mathrm{E}_s/\delta t$ at $T_t = 2$

(c) $\delta \mathbf{M}_s/\delta t$ at $T_t = 5$

(d) $\delta \mathrm{E}_s/\delta t$ at $T_t = 5$

(e) $\delta \mathbf{M}_s/\delta t$ at $T_t = 10$

(f) $\delta \mathrm{E}_s/\delta t$ at $T_t = 10$

**Figure 4.** The momentum and energy collision terms for the Maxwellian velocity distribution function in the case of Coulomb collisions, at different values of $T_t$: 2, 5, and 10.

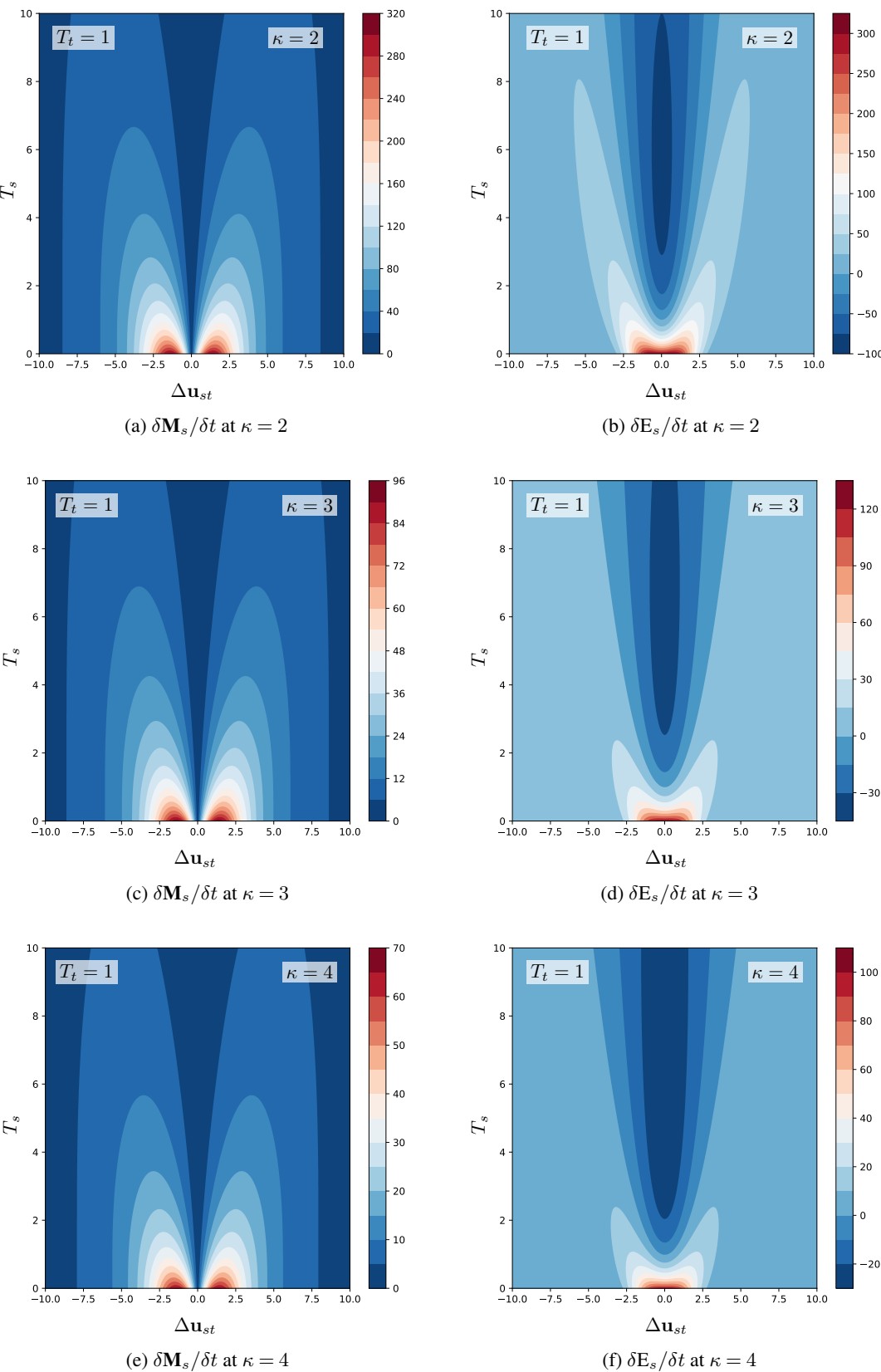

**Figure 5.** The momentum and energy collision terms for the modified Kappa velocity distribution function in the case of Coulomb collisions at different values of $\kappa$: 2, 3, and 4.

The change in the energy comes from the potential energy, specifically the difference between the temperatures of the interacting particles. Figure (3d) shows the cross-section of the energy collision term for the case $\Delta\mathbf{u}_{st} = 0$. At $T_s = 0$, the energy exchange between particles is most significant, as particles $s$ gain temperature from particles $t$. As $T_s$ increases and approaches $T_t$, the energy transfer decreases and vanishes when the temperatures of the two species are equal. This occurs at $T_s = T_t = 1$ in Figure (3d), where $\Delta\mathrm{T}_{st} = 0$, leading to no energy transfer and the energy collision term becomes zero. As the temperature of particles $s$ increases above the temperature of particles $t$, $T_s > T_t$, particles $s$ lose temperature to particles $t$, and the energy collision term becomes negative and decreases until it reaches a minimum value, which in our case occurs at $T_s = 5$. Beyond this point, the increase in the temperature of particles $s$ reduces the effective collision frequency, see equation (79). As a result, particles $s$ keep their temperature without losing any of it to particles $t$. This explains the subsequent increase in the change of energy. Eventually, as the effective collision frequency tends to zero, no further collisions occur, and the temperature change vanishes, making the energy collision term approach zero as $T_s$ goes to infinity. Back to Figure (3c), as the absolute value of the difference in drift velocity increases, the distance between particles also increases. This greater distance between particles makes collisions less likely, which in turn reduces energy exchange for species $s$. Consequently, the energy collision term approaches zero as the drift velocity difference, $\Delta\mathbf{u}_{st}$, tends to $\pm\infty$, similar to the behaviour of the momentum collision term when either $\mathbf{u}_s \gg \mathbf{u}_t$ or $\mathbf{u}_t \gg \mathbf{u}_s$.

**Third case**, the *temperature* of the $t$ particles $T_t$. In Figure (4), we plot the isolines of the collision terms under the same conditions as in Figure (3), but for different values of $T_t$. As $T_t$ increases, the collision terms exhibit the same overall behaviour, but their magnitude decreases. For example, $\delta\mathbf{M}_s/\delta t$ drops from 35 at $T_t = 1$ (Figure 3a) to 18 at $T_t = 2$ (Figure 4a), 7 at $T_t = 5$ (Figure 4c), and 3.5 at $T_t = 10$ (Figure 4e). A similar trend is observed for $\delta E_s/\delta t$, which drops from 60 to 48, then to 30, and finally to 21 (Figures 3c, 4b, 4d, and 4f). This reduction occurs because, as the temperature increases, the number of collisions decreases as mentioned before. Additionally, Figure (4) shows that the range of $\Delta\mathbf{u}_{st}$ contributing to the collision terms (red area along the horizontal axis) expands with $T_t$, since fewer collisions allow particles to accelerate more under external forces, increasing their drift velocities.

### 2.4.2 Modified Kappa distribution

The modified Kappa distribution affects the collision terms through two functions $\mathrm{D}(\kappa_s, \kappa_t)$ and $\mathrm{H}(\kappa_\alpha)$, $\alpha = s, t$, which appear in the effective collision frequency, the thermalisation rate and the relative temperature difference. Assuming equal kappa values for both species, $s$ and $t$, $\kappa_s = \kappa_t = \kappa$, allows for a direct comparison with the Maxwellian case. To understand how modified Kappa distribution changes the collision terms, we plot the isolines of the momentum and energy collision terms as functions of $\Delta\mathbf{u}_{st}$ and $T_s$, as shown in Figure (5), with the same conditions as in Figure (3a), for various $\kappa$ values. Cross-sections at $T_s = 0$ are shown in Figure 6. For the momentum collision term, the behavior is similar to the Maxwellian case, with $\mathrm{D}(\kappa, \kappa)$ scaling the effective collision frequency, as shown in Figures (5) and (6). At low $\kappa$, the effective collision frequency increases, as discussed in Section 2.3, leading to greater momentum transfer due to collisions at these values. For the energy collision term, the function $\mathrm{W}(\kappa, \kappa)$, defined as

$$\mathrm{W}(\kappa, \kappa) = \mathrm{D}(\kappa, \kappa)\,\mathrm{H}(\kappa), \tag{80}$$

appears in the first term of equation (42), while $\mathrm{D}(\kappa, \kappa)$ contributes to the second term. The general behavior of the energy coefficient is approximately the same as in the Maxwellian case, particularly at high $\kappa$ values. However, at low kappa values, particularly $\kappa = 2$, $\mathrm{D}(\kappa, \kappa) > \mathrm{W}(\kappa, \kappa)$, as outlined in Table (1), making the kinetic term dominant and producing peaks near zero, as shown in Figure 6b. Overall, both collision terms decrease with increasing $\kappa$, converging toward the Maxwellian result, confirming the result of Subsubsection 2.2.2.

**Table 1.** $\mathrm{D}(\kappa, \kappa)$ and $\mathrm{W}(\kappa, \kappa)$ at different values of $\kappa$.

| $\kappa$ | $\mathrm{D}(\kappa, \kappa)$ | $\mathrm{W}(\kappa, \kappa)$ |
| --- | --- | --- |
| 2 | 9.0000 | 4.7873 |
| 3 | 2.7778 | 2.0474 |
| 4 | 1.9600 | 1.5986 |
| 10 | 1.2491 | 1.1661 |
| 100 | 1.0204 | 1.0140 |
| 1000 | 1.0020 | 1.0014 |

## 3 Transport coefficients

In this section, we derive and discuss the behavior of the transport coefficients—namely, the electrical conductivity $\sigma_e$, thermoelectric coefficient $\alpha_e$, diffusion coefficient $D_e$, and mobility coefficient $\mu_e$—for a Lorentz plasma using the modified Kappa distribution. A Lorentz plasma is a type of plasma in which the contribution of electron-electron collisions is negligible compared to electron-ion collisions. In this model, electrons are considered to move relative to nearly stationary ions because their much smaller mass allows them to move much faster (Du, 2013). The Lorentz plasma model is particularly useful for calculating transport coefficients, such as electrical conductivity, because electron-electron collisions do not contribute significantly to these properties.

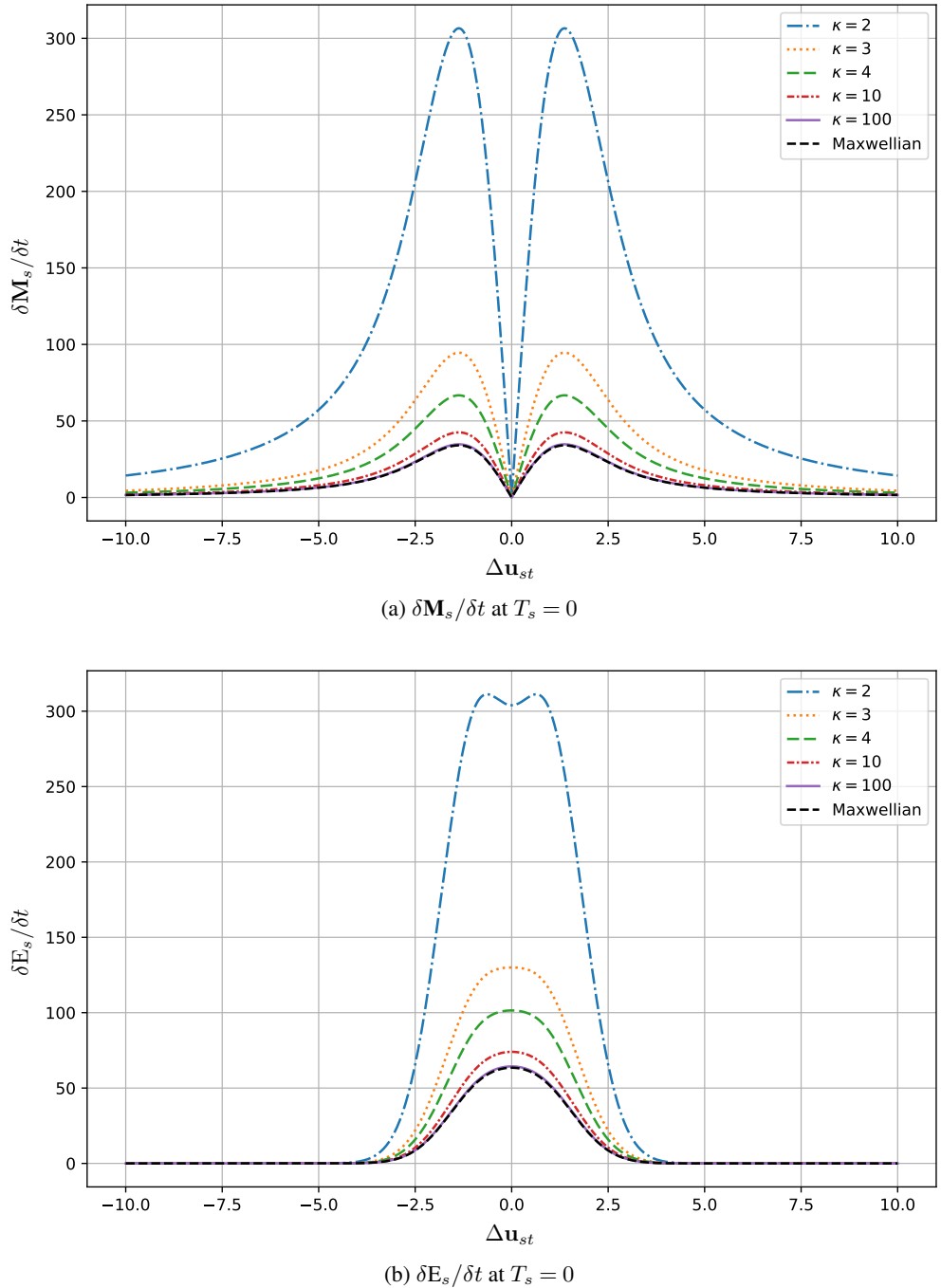

(a) $\delta \mathbf{M}_s / \delta t$ at $T_s = 0$

(b) $\delta \mathrm{E}_s / \delta t$ at $T_s = 0$

**Figure 6.** The cross-section of the momentum and energy collision terms for the modified Kappa and Maxwellian velocity distribution functions in the case of Coulomb collisions at $T_s = 0$.

The transport coefficients for Lorentz plasma without a magnetic field appear in the following macroscopic laws (Husidic et al., 2021),

$$\mathbf{E} = \frac{\mathbf{J}_e}{\sigma_e} + \alpha_e \nabla T_e, \tag{81}$$

$$\mathbf{\Gamma}_e = -D_e \nabla n_e - \mu_e n_e \mathbf{E}. \tag{82}$$

Equation (81) represents generalizes Ohm's law, where $\mathbf{E}$ denotes the electric field and $\mathbf{J}_e$ is the current density, defined as (Schunk and Nagy, 2009),

$$\mathbf{J}_e = e n_e (\mathbf{u}_i - \mathbf{u}_e), \tag{83}$$

with $e$ being the charge, $n_e$ the electron density, and $\mathbf{u}_i, \mathbf{u}_e$ the ion and electron velocities, respectively. Similarly, Equa-

tion (82) extends Fick's law to account for electric field effects, with $\boldsymbol{\Gamma}_e$ denotes the particle flux density, defined as (Schunk and Nagy, 2009),

$$\boldsymbol{\Gamma}_e = n_s \mathbf{u}_e. \tag{84}$$

## 3.1 Derivation of transport coefficients

The transport coefficients can be obtained by deriving equations (81) and (82) using the five-moment approximation. For a simple electron-ion collision, using equation (41), the momentum equation with a drifting modified Kappa distribution, can be expressed as

$$\frac{\delta \mathbf{M}_e}{\delta t} = n_e m_e \frac{\mathrm{D}_e \mathbf{u}_e}{\mathrm{D}t} + \nabla p_e$$
$$- n_e m_e \mathbf{G} + n_e e \left( \mathbf{E} + \frac{\mathbf{u}_e \times \mathbf{B}}{c} \right), \tag{85}$$

with the momentum collision term is given by

$$\frac{\delta \mathbf{M}_e}{\delta t} = n_e m_e \nu_{ei}^{\mathrm{MK}} \Delta \mathbf{u}_{st}, \tag{86}$$

where $\nu_{ei}^{\mathrm{MK}}$ is the effective collision frequency for the modified Kappa distribution, as defined in equation (47). In equation (85), the electron drift velocity is assumed negligible compared to the thermal velocity, which is equivalent to setting $\epsilon_{ei} = 0$ and therefore $\Phi(0) = 1$ in equation (41). To obtain the transport coefficients, we adopt the standard approximations. First, we assume a steady and low-inertia regime so that $n_e m_e D_e \mathbf{u}_e / Dt \approx 0$. Second, we neglect external gravity and magnetic fields, i.e., $\mathbf{G} = 0$ and $\mathbf{B} = 0$, which corresponds to unmagnetized scalar transport. For convenience, we also take the background ion flow to be $\mathbf{u}_i \approx 0$. Under these assumptions, the electron momentum equation reduces to

$$-n_e \mathbf{u}_e = \frac{k_B T_e}{m_e \nu_{ei}^{\mathrm{MK}}} \nabla n_e + \frac{n_e k_B}{m_e \nu_{ei}^{\mathrm{MK}}} \nabla T_e + \frac{n_e e}{m_e \nu_{ei}^{\mathrm{MK}}} \mathbf{E}, \tag{87}$$

since

$$\nabla p_e = k_B T_e \nabla n_e + n_e k_B \nabla T_e. \tag{88}$$

To derive the electrical conductivity and the thermoelectric coefficients, we begin by setting $\nabla n_e = 0$ and using the definition of the current density $\mathbf{J}_e$ with $\mathbf{u}_i \approx 0$, we write

$$\mathbf{J}_e = e n_e (\mathbf{u}_i - \mathbf{u}_e) = -e n_e \mathbf{u}_e. \tag{89}$$

Substituting this expression into the electron momentum equation (87), we obtain

$$\mathbf{J}_e = \frac{e n_e k_B}{m_e \nu_{ei}^{\mathrm{MK}}} \nabla T_e + \frac{n_e e^2}{m_e \nu_{ei}^{\mathrm{MK}}} \mathbf{E}. \tag{90}$$

Solving the equation for the electric filed gives

$$\mathbf{E} = \frac{m_e \nu_{ei}^{\mathrm{MK}}}{n_e e^2} \mathbf{J}_e - \frac{k_B}{e} \nabla T_e. \tag{91}$$

By comparing this result with the generalized Ohm's law, equation (81), we can identify the electrical conductivity and the thermoelectric coefficient as

$$\sigma_e = \frac{n_e e^2}{m_e \nu_{ei}^{\mathrm{MK}}}, \tag{92}$$

$$\alpha_e = -\frac{k_B}{e}. \tag{93}$$

Similar to how we derive the electrical conductivity and the thermoelectric coefficient, the diffusion and mobility coefficients can be obtained by setting $\nabla T_e = 0$ and using the definition of the particle flux density, $\boldsymbol{\Gamma}_e$, in equation (84), equation (87) gives

$$\boldsymbol{\Gamma}_e = -\frac{k_B T_e}{m_e \nu_{ei}^{\mathrm{MK}}} \nabla n_e - \frac{n_e e}{m_e \nu_{ei}^{\mathrm{MK}}} \mathbf{E}. \tag{94}$$

By comparing this to Fick's law, equation (82), the diffusion and mobility coefficients are identified as

$$D_e = \frac{k_B T_e}{m_e \nu_{ei}^{\mathrm{MK}}}, \tag{95}$$

$$\mu_e = \frac{e}{m_e \nu_{ei}^{\mathrm{MK}}}. \tag{96}$$

## 3.2 Discussion of transport coefficients

From the first look at the derived transport coefficients, we can see that they satisfy the familiar relation between the electric conductivity and the mobility coefficient

$$\sigma_e = n_e e \mu_e, \tag{97}$$

and Einstein relation

$$D_e = \frac{k_B T_e}{e} \mu_e. \tag{98}$$

The resulting transport coefficients show a different dependency on the kappa parameters. The thermoelectric coefficient $\alpha_e$ has no kappa term in it. On the other hand, the electrical conductivity, diffusion, and mobility coefficients all include the same kappa term, which appears through the effective collision frequency $v_{ei}^{\mathrm{MK}}$. The transport coefficients-electrical conductivity, diffusion, and mobility-are inversely proportional to the effective collision frequency. As discussed earlier, when $\kappa = \kappa_s = \kappa_t$, the effective collision frequency for the modified Kappa distribution influences different types of collisions in distinct ways. Consequently, the impact of the modified Kappa distribution on the transport coefficients depends on the specific type of collision. For Maxwell molecules, the effective collision frequency remains identical to that of the Maxwellian distribution, implying that the modified Kappa distribution does not affect the transport coefficients in this kind of collision. However, for Coulomb collisions and hard-sphere interactions, the effective collision frequency increases as $\kappa$ decreases, leading

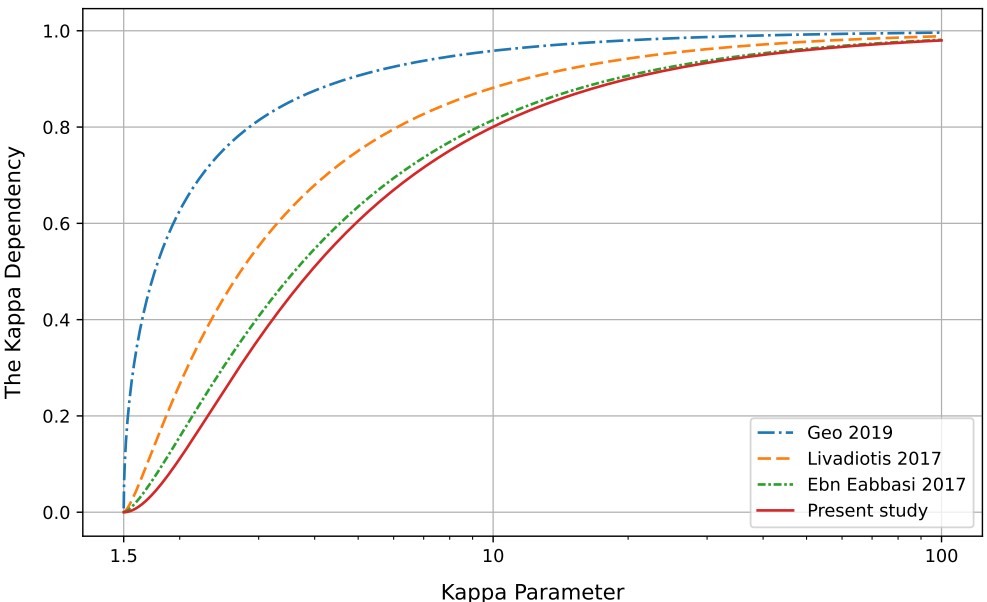

**Figure 7.** The kappa dependency for the electrical conductivity

to a decrease in the transport coefficients at low values of $\kappa$ compared to the Maxwellian case, as shown in Figure 7, where we have plotted the kappa dependency for the electrical conductivity as functions of the kappa parameter. As $\kappa$ approaches $\infty$, the effective collision frequency $v_{ei}^{MK}$ reduces to the Maxwellian case $v_{ei}$, making the transport coefficients recover their Maxwellian limits. The obtained transport coefficients have both differences and similarities with other studies. One can compare this with a number of studies that predicted a similar trend in the transport coefficients. In Figure 7, we also show the dependence on the $\kappa$ parameter of the electrical conductivity from different studies and compare it with the present work, by plotting the ratio $\sigma_e/\sigma_e^M$ as a function of $\kappa$, where $\kappa = \kappa_t = \kappa_s$, and

$$\sigma_e^M = \frac{n_e e^2}{m_e \nu_{ei}}. \tag{99}$$

All studies show a different dependence on the $\kappa$ parameter, but they still exhibit the same behavior: at low values of $\kappa$, the electrical conductivity becomes smaller compared to the Maxwellian case, and as $\kappa$ increases, we approach the Maxwellian case but never exceed it. This confirms that plasmas with larger $\kappa$ values are better conductors. Thus, deviations from the Maxwellian limit lead to a decrease in electrical conductivity. Figure 7 also shows that, the curves converge to the present work. This can be explained through the collision models used in the derivation of the transport coefficients. In each study, a more general collision term model was used. For example, Guo and Du (2019) used the Linearized Lorentz Collision Model with a relaxation-time approximation for electron-ion collisions. Livadiotis (2017)

used the Fokker–Planck collision operator with simplifications related to the direction of the electric field. Finally, the work of Ebne Abbasi et al. (2017) used the Fokker–Planck collision operator, where the derivation was generalized, and the only approximation occurred in the last step for the hypergeometric function. The work of Ebne Abbasi et al. (2017) is the closest to our present study, as can be seen clearly in the figure. The reason is that, in our study, we used the full Boltzmann collision integral, of which the Fokker–Planck operator can be considered an approximation.

## 4  Conclusions

For a Lorentz plasma described by a modified Kappa distribution, we have derived the transport coefficients: electrical conductivity, thermoelectric, diffusion, and mobility. The derivation begins with deriving a closed system of transport equations for isotropic plasmas within the five-moment approximation. In these equations, the transport properties of a given species are defined with respect to the random velocity of that species, where the species velocity distribution function is expanded in an orthogonal polynomial series about a drifting modified Kappa weighting function. By taking only the first term of the expansion and neglecting all higher order moments of the velocity distribution, we obtain the five-moment approximation. The corresponding momentum and energy collision terms were evaluated via the Boltzmann collision integral for several interaction types, including Coulomb collisions, hard-sphere interactions, and Maxwell molecule collisions. Given the complexity of the collision terms, the final expressions were represented us-

ing hypergeometric functions to simplify numerical calculations. Next, we investigated the limiting case in which the kappa index approaches infinity, where the collision terms reduce to Maxwellian form. Then we analyzed the influence of the kappa index on the effective collision frequency and thermalisation rate. It was observed that systems described by modified Kappa distribution exhibit distinct behaviour compared to those with Maxwellian distribution. The modified kappa distribution directly influences collision dynamics, where, for interactions in which collision frequency is velocity-independent, such as Maxwell molecule interaction, the distribution does not affect the effective collision frequency or thermalisation rate. In contrast, for velocity-dependent collisions, such as Coulomb and hard-sphere interactions, smaller $\kappa$ values increase collision frequency and thermalization, and larger $\kappa$ values approach the Maxwellian limit. We further examined the influence of the kappa index on the momentum and energy collision terms of the particles during Coulomb collisions. At low kappa parameter values, the number of collisions increases significantly, making both the effective collision frequency and the thermalisation rate produce greater changes in the momentum and energy of the particles, while larger $\kappa$ values recover Maxwellian behavior. Starting from the momentum equation and applying suitable assumptions for an unmagnetized, steady-state plasma, explicit expressions for the electrical conductivity, thermoelectric coefficient, diffusion coefficient, and mobility coefficient were obtained for the modified Kappa distribution. The analysis reveals that while the thermoelectric coefficient is unaffected by the $\kappa$ parameter, the electrical conductivity, diffusion, and mobility coefficients are all inversely proportional to the effective collision frequency, thereby reflecting its $\kappa$ dependency. Furthermore, the results indicate that lower $\kappa$ values lead to an increase in collision frequency and consequently a decrease in the transport coefficients. In the limit ($\kappa \to \infty$), the coefficients naturally reduce to Maxwellian form, confirming the consistency of the approach.

While the current study provides an important step towards a comprehensive non-Maxwellian transport theory, the present work is limited in several ways. First, the approach was derived within the five-moment approximation of the transport equation, considering only the first term of the expansion and neglecting higher-order moments. This simplification represents a poor approximation, as the neglected terms could significantly affect the system's behavior. Second, the analysis assumes isotropic plasmas, which restricts its applicability to real space plasma environments. In reality, most space plasmas are magnetized and exhibit temperature and pressure anisotropies. Ignoring these effects may overlook important physical mechanisms that govern plasma dynamics. Third, the Coulomb collision cross-section was simplified using a constant Coulomb logarithm and large-velocity approximation, which may affect quantitative accuracy at low velocities (Fichtner et al., 1996). Finally, the model employs the modified Kappa distribution, whose applicability breaks down for $\kappa \leq 3/2$, as lower values cause divergent moments and thermodynamic inconsistencies. In this regime, the functions $D(\kappa, \kappa)$ and $H(\kappa)$ diverge, making the effective collision frequency and the thermalisation rate become unphysical. Therefore, the derived collision terms and transport coefficients are valid only for $\kappa > 3/2$.

Future work should address the current model's limitations through several key extensions. First, developing a comprehensive transport theory that accounts for the modified Kappa velocity distribution, would significantly advance the framework beyond the standard Maxwellian assumption. This can be achieved by expanding the species distribution function in a generalized orthogonal polynomial series with a $\kappa$-weighting function, allowing for systematic derivation of various approximations (e.g. eight-, ten-, thirteen-, and twenty-moment models). Second, extending the theory to anisotropic plasmas—where pressure and temperature vary with direction—would enhance its realism and applicability, particularly in magnetized and space plasma environments. In addition, incorporating the exact velocity-dependent Coulomb cross-section would improve the accuracy of the collision transfer integrals. Furthermore, adopting the Regularized Kappa Distribution proposed by (Scherer et al., 2017, 2019), which preserves the core features of the $\kappa$-function while ensuring finite moments for all $\kappa > 0$ and maintaining thermodynamic consistency, would provide a more stable and physically meaningful representation.

## Appendix A

This appendix presents the derivation of the collision terms for each type of collision for the drifting modified Kappa distribution introduced in Section 2.2.

### A1   Coulomb collisions

The collision terms for Coulomb collisions can be obtained by substituting equation (37) into equations (32-34), leading to the following form

$$\frac{\delta n_s}{\delta t} = 0, \tag{A1}$$

$$\frac{\delta \mathbf{M}_s}{\delta t} = -\sum_t m_{st} \, Q_{\text{Co}} \int_{\mathbb{R}^3 \times \mathbb{R}^3} f_s^{\text{MK}} f_t^{\text{MK}} \frac{\mathbf{g}_{st}}{\mathbf{g}_{st}^3} \, d\mathbf{c}_t \, d\mathbf{c}_s, \tag{A2}$$

$$\frac{\delta E_s}{\delta t} = -\sum_t m_{st} \, Q_{\text{Co}} \int_{\mathbb{R}^3 \times \mathbb{R}^3} f_s^{\text{MK}} f_t^{\text{MK}} \frac{(\hat{\mathbf{V}}_c \cdot \mathbf{g}_{st})}{\mathbf{g}_{st}^3} \, d\mathbf{c}_t \, d\mathbf{c}_s. \tag{A3}$$

*The Momentum Coefficient*

The momentum coefficient for Coulomb collision is given in

equation (A2). Evaluating it requires computing the integral

$$I_M = \int\limits_{\mathbb{R}^3 \times \mathbb{R}^3} f_s^{\mathrm{MK}} f_t^{\mathrm{MK}} \frac{\mathbf{g}_{st}}{\mathrm{g}_{st}^3} \, d\mathbf{c}_t \, d\mathbf{c}_s, \tag{A4}$$

to do this, we begin by rewriting the distribution function of the interacting species $s$ and $t$ in an integral representation, instead of writing them in the form of equation (20), as

$$f_s^{\mathrm{MK}} = \frac{n_s \eta(\kappa_s)}{\pi^{3/2} \, w_s^3} \frac{1}{\Gamma(\kappa_s + 1)} \int_0^\infty \xi_1^{\kappa_s} \, e^{-\xi_1} \exp\left[-\frac{\xi_1 \, c_s^2}{\kappa_{0_s} \, w_s^2}\right] d\xi_1, \tag{A5}$$

$$f_t^{\mathrm{MK}} = \frac{n_t \eta(\kappa_t)}{\pi^{3/2} \, w_t^3} \frac{1}{\Gamma(\kappa_t + 1)} \int_0^\infty \xi_2^{\kappa_t} \, e^{-\xi_2} \exp\left[-\frac{\xi_2 \, c_t^2}{\kappa_{0_t} \, w_t^2}\right] d\xi_2. \tag{A6}$$

Substituting the distributions given in equations (A5) and (A6) in the integral, equation (A4), gives

$$I_M = \frac{n_s n_t}{\pi^3 \, w_s^3 \, w_t^3} \frac{\eta(\kappa_s)\eta(\kappa_t)}{\Gamma(\kappa_s + 1)\Gamma(\kappa_t + 1)} \int_0^\infty \int_0^\infty \xi_1^{\kappa_s} \, \xi_2^{\kappa_t} \, e^{-\xi_1 - \xi_2}$$
$$\left( \int\limits_{\mathbb{R}^3} \int\limits_{\mathbb{R}^3} \exp\left[-\frac{\xi_1 \, c_s^2}{\kappa_{0_s} \, w_s^2} - \frac{\xi_2 \, c_t^2}{\kappa_{0_t} \, w_t^2}\right] \frac{\mathbf{g}_{st}}{\mathrm{g}_{st}^3} \, d\mathbf{c}_s d\mathbf{c}_t \right) d\xi_1 \, d\xi_2. \tag{A7}$$

To solve the integral in equation (A7), we introduce the following transformation

$$\mathbf{c}_s = \left(\frac{\kappa_{0_s}}{\xi_1}\right)^{1/2} (\mathbf{c}_* - \mathbf{A}\mathbf{g}_*), \tag{A8}$$

$$\mathbf{c}_t = \left(\frac{\kappa_{0_t}}{\xi_2}\right)^{1/2} (\mathbf{c}_* + \mathbf{B}\mathbf{g}_*). \tag{A9}$$

The constant A and B are defined by:

$$A = \frac{m_t T_s}{m_t T_s + m_s T_t}, \qquad B = \frac{m_s T_t}{m_t T_s + m_s T_t}. \tag{A10}$$

Here $\mathbf{c}_*$ and $\mathbf{g}_*$ are expressed as:

$$\mathbf{c}_* = \mathbf{V}_c - \mathbf{u}_c + \xi \, \Delta \mathbf{u}_{st} + \xi \, \mathbf{g}_{st}, \tag{A11}$$

$$\mathbf{g}_* = -\mathbf{g}_{st} - \Delta \mathbf{u}_{st}, \tag{A12}$$

with

$$\mathbf{u}_c = \frac{m_s \mathbf{u}_s + m_t \mathbf{u}_t}{m_s + m_t}, \qquad \xi = \frac{m_{st}}{m_s + m_t} \frac{T_t - T_s}{T_{st}}. \tag{A13}$$

Calculating the determinant of the Jacobian matrix **J** for the transformation described in equations (A8-A10) gives

$$d\mathbf{c}_s d\mathbf{c}_t = \det(\mathbf{J}) \, d\mathbf{c}_* d\mathbf{g}_* = \left[\frac{\kappa_{0_s} \kappa_{0_t}}{\xi_1 \xi_2}\right]^{1/2} d\mathbf{c}_* d\mathbf{g}_*. \tag{A14}$$

Applying the transformation make the integrals in equation (A7) independent of each other, so by evaluating the integrals with respect to $\mathbf{c}_*$, $\xi_1$ and $\xi_2$ , and rewriting the $\mathbf{g}_*$ integral, we have

$$I_M = \frac{n_s n_t}{\pi^{3/2} \, b^3} \, \mathrm{D}(\kappa_s, \kappa_t) \int\limits_{\mathbb{R}^3} e^{-|\mathbf{g}_{st} + \Delta\mathbf{u}_{st}|^2 / w_{st}^2} \, \frac{\mathbf{g}_{st}}{\mathrm{g}_{st}^3} \, d\mathbf{g}_{st}. \tag{A15}$$

where $\mathrm{D}(\kappa_s, \kappa_t)$ is defined in equation (52). The integral on $\mathbf{g}_{st}$ can be solved by setting $z$-axis in the direction of vector $\Delta\mathbf{u}_{st}$, so that the angle between $\mathbf{g}_{st}$ and $\Delta\mathbf{u}_{st}$ corresponds to the polar angle in spherical coordinates. We then transform the integrals into spherical coordinates and perform the integration, to get

$$I_M = -\frac{4}{3} \frac{n_s n_t}{\pi^{1/2} \, w_{st}^3} \, \mathrm{D}(\kappa_s, \kappa_t) \, \Phi_{\mathrm{Co}}(\varepsilon_{st}) \, \Delta\mathbf{u}_{st}. \tag{A16}$$

Substituting equation (A16) into equation (A2), we can write the momentum coefficient for Coulomb collision as

$$\frac{\delta \mathbf{M}_s}{\delta t} = \sum_t n_s m_s \nu_{st}^{\mathrm{Co}} \mathrm{D}(\kappa_s, \kappa_t) \Phi_{\mathrm{Co}}(\varepsilon_{st}) \, \Delta\mathbf{u}_{st}, \tag{A17}$$

where $\nu_{st}^{\mathrm{Co}}$ and $\Phi_{\mathrm{Co}}$ are defined in equations (49) and (50).

*The Energy Coefficient*

The energy coefficient for Coulomb collisions is given in equation (A3). Substituting the dot product from equation (35) into the integral

$$I_E = \int\limits_{\mathbb{R}^3 \times \mathbb{R}^3} f_s^{\mathrm{MK}} f_t^{\mathrm{MK}} \frac{(\hat{\mathbf{V}}_c \cdot \mathbf{g}_{st})}{\mathrm{g}_{st}^3} \, d\mathbf{c}_t \, d\mathbf{c}_s, \tag{A18}$$

produces three integrals, which can be evaluated by following the same steps as the integral in equation (A4), where we substitute the distributions given in equations (A5) and (A6) into the integrals, and apply the transformation mentioned in equations (A8-A10). This results in two integrals that depend on $\mathbf{g}_{st}$, which can be evaluated using the same steps used for the integral in equation (A15). Combining all integrals, we obtain:

$$I_E = -\frac{4}{3} \frac{1}{m_s + m_t} \frac{n_s n_t}{\pi^{1/2} \, w_{st}^3} \left[3 k_B \, \mathrm{D}(\kappa_s, \kappa_t) \, \Psi_{\mathrm{Co}}(\varepsilon_{st}) \, \Delta\mathrm{T}_{st}^{\mathrm{MK}} \right.$$
$$\left. + m_t \, \mathrm{D}(\kappa_s, \kappa_t) \, \Phi_{\mathrm{Co}}(\varepsilon_{st}) \, |\Delta\mathbf{u}_{st}|^2 \right], \tag{A19}$$

Substituting equation (A19) into equation (A3), we can write the energy coefficient for Coulomb collision as

$$\frac{\delta E_s}{\delta t} = \sum_t \frac{n_s m_s \nu_{st}^{\mathrm{Co}}}{m_s + m_t} \left[3 k_B \, \mathrm{D}(\kappa_s, \kappa_t) \, \Psi_{\mathrm{Co}}(\varepsilon_{st}) \Delta\mathrm{T}_{st}^{\mathrm{MK}} \right.$$
$$\left. + m_t \, \mathrm{D}(\kappa_s, \kappa_t) \, \Phi_{\mathrm{Co}}(\varepsilon_{st}) \, |\Delta\mathbf{u}_{st}|^2 \right], \tag{A20}$$

where $\Delta\mathrm{T}_{st}^{\mathrm{MK}}$ and $\Psi_{\mathrm{Co}}$ are defined as in equations (44) and (51), respectively.

## A2 Hard-sphere interactions

The collision terms for hard-sphere interactions can be obtained by substituting equation (38) into equations (32-34), leading to the following form

$$\frac{\delta n_s}{\delta t} = 0, \tag{A21}$$

$$\frac{\delta \mathbf{M}_s}{\delta t} = -\sum_t m_{st} \mathrm{Q}_{\mathrm{HS}} \int_{\mathbb{R}^3 \times \mathbb{R}^3} f_s^{\mathrm{MK}} f_t^{\mathrm{MK}} \mathbf{g}_{st} \, \mathbf{g}_{st} \, d\mathbf{c}_t \, d\mathbf{c}_s, \tag{A22}$$

$$\frac{\delta E_s}{\delta t} = -\sum_t m_{st} \mathrm{Q}_{\mathrm{HS}} \int_{\mathbb{R}^3 \times \mathbb{R}^3} f_s^{\mathrm{MK}} f_t^{\mathrm{MK}} \mathbf{g}_{st} \left( \hat{\mathbf{V}}_c \cdot \mathbf{g}_{st} \right) d\mathbf{c}_t \, d\mathbf{c}_s. \tag{A23}$$

Calculating the collision terms for hard-sphere interactions follows the same steps as those applied previously for the Coulomb collision. The main difference between the integrals in equations (A21−A23) and (A1−A3) is that $g_{st}$ has a power of 1 rather than $-3$. This difference affects only the final integrals. Using the same technique, we obtain the momentum coefficient for hard-sphere interactions,

$$\frac{\delta \mathbf{M}_s}{\delta t} = \sum_t n_s m_s \nu_{st}^{\mathrm{HS}} \mathrm{D}(\kappa_s, \kappa_t) \, \Phi_{\mathrm{HS}}(\varepsilon_{st}) \, \Delta \mathbf{u}_{st}, \tag{A24}$$

and the energy coefficient for hard-sphere interactions,

$$\frac{\delta E_s}{\delta t} = \sum_t \frac{n_s m_s \nu_{st}^{\mathrm{HS}}}{m_s + m_t} \left[ 3 k_B \, \mathrm{D}(\kappa_s, \kappa_t) \, \Psi_{\mathrm{HS}}(\varepsilon_{st}) \, \Delta \mathrm{T}_{st}^{\mathrm{MK}} \right.$$
$$\left. + m_t \, \mathrm{D}(\kappa_s, \kappa_t) \, \Phi_{\mathrm{HS}}(\varepsilon_{st}) \, |\Delta \mathbf{u}_{st}|^2 \right], \tag{A25}$$

where $\nu_{st}^{\mathrm{HS}}$, $\Phi_{\mathrm{HS}}$, and $\Psi_{\mathrm{HS}}$ are defined in equations (54-56).

## A3 Maxwell molecule collisions

The collision terms for Maxwell molecule collisions can be obtained by substituting equation (39) into equations (32-34), leading to the following form

$$\frac{\delta n_s}{\delta t} = 0, \tag{A26}$$

$$\frac{\delta \mathbf{M}_s}{\delta t} = -\sum_t m_{st} \mathrm{Q}_{\mathrm{MC}} \int_{\mathbb{R}^3 \times \mathbb{R}^3} f_s^{\mathrm{MK}} f_t^{\mathrm{MK}} \mathbf{g}_{st} \, d\mathbf{c}_t \, d\mathbf{c}_s, \tag{A27}$$

$$\frac{\delta E_s}{\delta t} = -\sum_t m_{st} \mathrm{Q}_{\mathrm{MC}} \int_{\mathbb{R}^3 \times \mathbb{R}^3} f_s^{\mathrm{MK}} f_t^{\mathrm{MK}} \left( \hat{\mathbf{V}}_c \cdot \mathbf{g}_{st} \right) d\mathbf{c}_t \, d\mathbf{c}_s. \tag{A28}$$

There's no integration technique required to evaluate equations (A26−A28), since after rewriting the relative velocity in equation (4) and the dot product in equation (35) in terms

of the random velocities $\mathbf{c}_s$ and $\mathbf{c}_t$, allows us to use the following expectation values for the modified Kappa distribution,

$$\left\langle c_\alpha^2 \right\rangle = \frac{3 k_B T_\alpha}{m_\alpha}, \quad \left\langle \mathbf{c}_\alpha \right\rangle = 0, \quad \left\langle A \right\rangle = A, \tag{A29}$$

where $A$ is constant and $\alpha$ denotes the species type $s$ or $t$, to obtain the momentum coefficient for Maxwell molecule collisions

$$\frac{\delta \mathbf{M}_s}{\delta t} = \sum_t n_s m_s \nu_{st}^{\mathrm{MC}} \Delta \mathbf{u}_{st}, \tag{A30}$$

and the energy coefficient for Maxwell molecule collisions

$$\frac{\delta E_s}{\delta t} = \sum_t \frac{n_s m_s \nu_{st}^{\mathrm{MC}}}{m_s + m_t} \left[ 3 k_B \, \Delta \mathrm{T}_{st} + m_t \, |\Delta \mathbf{u}_{st}|^2 \right], \tag{A31}$$

where $\nu_{st}^{\mathrm{MC}}$ is defined in equation (57).

*Author contributions.* Mahmood Jwailes carried out the theoretical work, derived the framework used to obtain the figures and results, wrote the initial manuscript, and led the discussion of the findings. Imad Barghouthi proposed the research idea, supervised the study, verified the validity of the results, and assisted in scientific editing of the manuscript. Qusay Atawnah contributed to the final revisions of the manuscript.

*Competing interests.* The authors declare that they have no competing interests.

*Disclaimer.* The views expressed in this article are solely those of the authors and do not necessarily represent the views of their affiliated institutions.

*Acknowledgements.* The authors thank the reviewers, Dr. Horst Fichtner and Prof. Marina Stepanova, for their critical reading of the manuscript and for their constructive suggestions, which significantly improved the quality of this work.

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
