# Peer review of "Transport coefficients in modified Kappa distributed plasmas"

_EGUsphere, 2025_

## Author Comment (AC2)

**Response to Reviewer # 1 Comments**

We sincerely thank Dr. Horst Fichtner for the careful reading of our manuscript and for the constructive suggestions provided. We have rewritten the manuscript accordingly to address all points raised. Below, we detail how each comment has been handled.

**Major Comments**

**1. Relation to closely related literature (Du 2013):**

**2. Comparison to Husidic et al. (2021):**

A discussion has been added in highlighting both similarities and differences between our transport coefficient calculations and those obtained by Du and Husidic with the standard kappa distribution. And we have added a new section that relates the five-moment approximation equations to the transport coefficients: electrical conductivity, thermoelectric coefficient, diffusion coefficient, and mobility coefficient, by showing how we can go from the momentum equation to the generalization of Ohm's law and the extended Fick's law.

**3. Limitations of the standard kappa distribution and use of the RKD:**

We have included a discussion on the known limitations of the standard kappa distribution ($\kappa > 3/2$, diverging velocity moments), referencing Scherer et al. (2019, *Astrophys. J.*, 881:93). Additionally, we now mention the regularized kappa distribution (RKD) introduced by Scherer et al. (2017, *Europhys. Lett.*, 120, 50002) and its application in Husidic et al. (2022, *Astrophys. J.*, 927:159).

**Minor Comments**

**(a) Burgers (1969) reference:** We have corrected the reference to include the full publisher information.

**(b) Citation formatting:** All citations in the text have been checked and adjusted to appear in parentheses where appropriate.

**(c) Burgers' results for Maxwellian case:** We have added a note mentioning that the Burgers (1969) results for the Maxwellian case are approximations, referencing Fichtner et al. (1996, *J. Plasma Phys.*, 55, 95).

**(d) Magnetized space plasma:** We have mentioned Guo & Du (2019, Physica A, 523, 156) when we discussed Du and Husidic.

---

## Author Response (AR1)

Dear Editor

We sincerely thank you for handling our manuscript and for providing the opportunity to revise it. We also express our gratitude to the reviewers, Dr. Horst Fichtner and Prof. Marina Stepanova, for their careful evaluation and constructive feedback, which have significantly improved the quality and clarity of our work. In response to their comments, We have conducted a major revision of our manuscript. The main changes are summarized as follows:

- **Relation to previous literature:** We expanded the introduction to include discussions of transport coefficient derivations and their connections to the works of Du (2013) and Husidic et al. (2021). We also added a new section linking the five-moment approximation equations to key transport coefficients (electrical conductivity, thermoelectric, diffusion, and mobility coefficients) and compared our results with several related studies.

- **Distribution function discussion:** We included a detailed discussion on the limitations of the kappa distribution and mentioned (in the conclusion) the regularized kappa distribution (RKD), referencing Scherer et al. (2017, 2019).

- **Reorganization and clarity improvements:** Sections 2 and 3 were shortened by moving detailed derivations to the Appendix, allowing the main text to focus on physical interpretation and new insights.

- **Expanded Introduction and Conclusions:** The Introduction now better articulates the motivation, background, and research gap, while the Conclusions emphasize the main findings, limitations, and potential applications in space plasma modeling.

- **Technical corrections:** We added full publisher information for the Burgers (1969) reference, corrected citation formatting, clarified Burgers' Maxwellian results, and mentioned Guo & Du (2019) in the relevant discussion. Figures 1–3 now include labeled colorbars, and Figure 4 features distinct line styles to clearly differentiate overlapping cases.

We believe these revisions address all points raised and improve the overall presentation and impact of the paper. We hope the revised version meets the journal's standards for publication.

**Response to Reviewer # 1 Comments**

We sincerely thank Reviewer # 1 (Dr. Horst Fichtner) for the careful and critical reading of our manuscript and for his respected suggestions. We have conducted a major revision of our manuscript due to your respected constructive suggestions. Below, we detail how each comment has been handled.

**Major Comments**

**1. Relation to closely related literature (Du 2013):**

**2. Comparison to Husidic et al. (2021):**

A discussion has been added in the introduction highlighting literature on the transport coefficient derivations and those obtained by Du and Husidic. We have added a new section that relates the five-moment approximation equations to the transport coefficients: electrical conductivity, thermoelectric coefficient, diffusion coefficient, and mobility coefficient, by showing how we can go from the momentum equation to the generalization of Ohm's law and the extended Fick's law. Finally, we have added a comparison between our results and several other studies that are closely related to our work.

**3. Limitations of the kappa distribution and use of the RKD:**

We have included a discussion on the known limitations of the standard kappa distribution ($\kappa > 3/2$, diverging velocity moments), referencing Scherer et al. (2019, *Astrophys. J.*, 881:93). Additionally, we mention the regularized kappa distribution (RKD) introduced by Scherer et al. (2017, *Europhys. Lett.*, 120, 50002) at the end of our conclusions.

**Minor Comments**

**(a) Burgers (1969) reference:** We have corrected the reference to include the full publisher information.

**(b) Citation formatting:** All citations in the text have been checked and adjusted to appear in parentheses where appropriate.

**(c) Burgers' results for Maxwellian case:** We have added a note mentioning that the Burgers (1969) results for the Maxwellian case are approximations, referencing Fichtner et al. (1996, *J. Plasma Phys.*, 55, 95).

**(d) Magnetized space plasma:** We have mentioned Guo & Du (2019, Physica A, 523, 156) when we discussed Du and Husidic.

**Response to Reviewer #2 Comments**

We thank Reviewer #2 (Prof. Marina Stepanova) for her positive evaluation of our manuscript and for recommending it for publication after revisions. We have conducted a major revision of our manuscript due to your respected constructive suggestions, addressed each point as follows:

**Specific Comments**

**1. Introduction too brief; need better articulation of significance and prior work**

**Reviewer comment:** *The introduction is too brief... it is necessary to better articulate the significance... Specifically, what prior work has been done in this specific topic, what gap does this study aim to address?*
**Response:** We agree and have revised the Introduction to provide a clearer context for the study. Specifically:

- We now summarize prior works analyzing transport processes in Maxwellian plasmas and the few attempts made for Kappa-distributed plasmas.

- We highlight that previous studies largely focused on simplified models.

- We explicitly state the gaps and the aim of the study more clearly.

**2. Sections 2 and 3 are long; suggest moving equations to Appendix**

**Reviewer comment:** *Sections 2 and 3... suggest shortening by moving a significant portion of the equations to the Appendix.*
**Response:** We have substantially shortened Sections 2 and 3 by transferring several detailed derivations to the Appendix. The main text now focuses on the physical interpretation and novel aspects of our formulation.

**3. Expand Conclusions to emphasize new insights and applications**

**Reviewer comment:** *I would also suggest expanding the conclusions section to emphasize what new insights we have gained...*

**Response:** We have expanded the Conclusions section to:

- Highlight the main findings.

- Discuss a number of limitations and how we can address them in future work.

- Discuss the potential applications of these results in space plasma modeling.

**Technical Corrections**

**4. Add labels near colorbars (Figures 1–3)**

**Response:** We have revised Figures 1–3 to include clear labels near the colorbars. Temperature and kappa values are now indicated within the plots to improve clarity.

**5. Differentiate overlapping lines in Figure 4 (kappa = 100, 1000 vs Maxwellian)**

**Response:** We have modified Figure 4 to use dotted and dash-dotted line styles for $\kappa = 100$ and $\kappa = 1000$, respectively. This makes it easier to visually distinguish these cases from the Maxwellian result.